# Datasets and Benchmarks for Nanophotonic Structure and Parametric Design Simulations

**Jungtaek Kim**
University of Pittsburgh
Pittsburgh, PA 15261, USA
jungtaek.kim@pitt.edu

**Mingxuan Li**
University of Pittsburgh
Pittsburgh, PA 15261, USA
mil152@pitt.edu

**Oliver Hinder**
University of Pittsburgh
Pittsburgh, PA 15261, USA
ohinder@pitt.edu

**Paul W. Leu**
University of Pittsburgh
Pittsburgh, PA 15261, USA
pleu@pitt.edu

## Abstract

Nanophotonic structures have versatile applications including solar cells, anti-reflective coatings, electromagnetic interference shielding, optical filters, and light emitting diodes. To design and understand these nanophotonic structures, electrodynamic simulations are essential. These simulations enable us to model electromagnetic fields over time and calculate optical properties. In this work, we introduce frameworks and benchmarks to evaluate nanophotonic structures in the context of parametric structure design problems. The benchmarks are instrumental in assessing the performance of optimization algorithms and identifying an optimal structure based on target optical properties. Moreover, we explore the impact of varying grid sizes in electrodynamic simulations, shedding light on how evaluation fidelity can be strategically leveraged in enhancing structure designs.

## 1 Introduction

Nanophotonic structures play a crucial role for a wide range of real-world applications such as solar cells, anti-reflective coatings, electromagnetic interference shielding, optical filters, and light emitting diodes.[1] In particular, several studies have demonstrated that nanophotonic structures can enhance target performance for many applications [70, 71, 73, 21, 19, 39]. Electrodynamic simulations, which are based on Maxwell's equations, provide accurate predictions of the optical and electromagnetic properties of these structures [18]. Leveraging the utility of these simulations, we can combine them with optimization procedures to design and discover new and improved structures.

In this paper, we introduce datasets and benchmarks for nanophotonic structures, focusing on parametric design simulations in relation to optical properties. These benchmarks allow us to analyze the performance of optimization algorithms such as derivative-free algorithms [53] and Bayesian optimization [16], when it comes to identifying optimal photonic structures based on a target optical property. Our frameworks support two modes for fast prototyping of optimization methods: discretized search space mode and surrogate model mode. The discretized search space mode uses stored simulated results over grid query points; the surrogate model mode estimates any query points within a continuous search space based on training with our datasets. These modes allow users to quickly test their optimization algorithms without running expensive electrodynamic simulations. Our system facilitates the use of evaluation fidelity, which is controlled by a simulation

---

[1]Our implementation can be found at https://github.com/jungtaekkim/nanophotonic-structures.

37th Conference on Neural Information Processing Systems (NeurIPS 2023) Track on Datasets and Benchmarks.

resolution, and the handling of multiple objectives. Moreover, our frameworks can also provide electromagnetic fields as a function of position and time.

Our contributions are summarized as follows:

- Development of a generic simulation scheme and pipeline for nanophotonic structures in Python, based on the open-source software, Meep, and licensed under the MIT license;
- Creation of datasets of a myriad of nanophotonic structures for electromagnetic interference shielding, anti-reflection, and solar cells;
- Investigation into the effects of altering grid sizes in electrodynamic simulations, providing insights into tradeoffs between computational time and simulation accuracy;
- Introduction of benchmarks specifically designed for the optimization of parametric structures, facilitating the evaluation and comparison of different optimization algorithms.

## 2 Background

In this section, we delve into the optical and electromagnetic properties of materials, explore nanophotonic structure designs, and review related literature.

### 2.1 Optical and Electromagnetic Properties of Materials

The optical and electromagnetic properties of materials can be determined by solving Maxwell's equations [18]. When studying materials featuring sizes smaller than wavelengths of interest, geometrical optics becomes unsuitable, and ray-tracing methods are inaccurate [7]. In such cases, it is crucial to use a simulation method that captures the wave-like nature of light to accurately address phenomena like interference and diffraction. Classical electromagnetic theory can predict reflection, absorption, and transmission spectra by solving Maxwell's equations:

$$\nabla \cdot \boldsymbol{E} = \frac{\rho}{\epsilon_0}, \qquad \nabla \cdot \boldsymbol{B} = 0, \qquad \nabla \times \boldsymbol{E} = -\frac{\partial \boldsymbol{B}}{\partial t}, \qquad \nabla \times \boldsymbol{B} = \mu_0 \boldsymbol{J} + \mu_0 \epsilon_0 \frac{\partial \boldsymbol{B}}{\partial t}, \quad (1)$$

where $\boldsymbol{E}$ and $\boldsymbol{B}$ are electric and magnetic fields, $\rho$ and $\boldsymbol{J}$ are charge and current densities, and $\epsilon_0$ and $\mu_0$ are the permittivity and permeability of free space. The speed of light is defined as $c = 1/\sqrt{\mu_0 \epsilon_0}$.

As shown in Figure S1, light consists of two synchronized waves: *electric fields* and *magnetic fields*. These fields are characterized by an angular frequency $\omega$, which can also be expressed as a frequency $f$, energy $E$, or wavelength $\lambda$, with the relationships $\omega = 2\pi f$, $E = hf$, and $E = hc/\lambda$ where $h$ is the Planck constant (i.e., $6.63 \times 10^{-34}$ J-sec) and $c$ is the speed of light (i.e., $3 \times 10^8$ m/sec). The interaction of light with materials is characterized by either a complex refractive index, $n(\omega) = n_r(\omega) + n_i(\omega)$, or a complex permittivity, $\epsilon(\omega) = \epsilon_r(\omega) + \epsilon_i(\omega)$, both of which describe how light bends and is absorbed within that material. The relative permittivities of the materials used in this paper are detailed in Section E. An important aspect of light-material interactions is understanding the transmission, reflection, and absorption spectra from a specific light source as it interacts with a material at a given frequency, as depicted in Figure S2. As governed by the law of conservation of energy [18], the following equation is satisfied:

$$R(\omega) + A(\omega) + T(\omega) = 1, \qquad (2)$$

where $R(\omega)$, $A(\omega)$, and $T(\omega)$ are the reflectance, absorbance, and transmittance of a material at a particular frequency $\omega$.

### 2.2 Designs of Nanophotonic Structures for Optoelectronic Applications

Numerous electrodynamic simulation techniques such as the transfer matrix method [6], finite element method [29], rigorous coupled-wave analysis [43], and finite-difference time-domain (FDTD) method [65] are important for nanophotonics research. FDTD, in particular, is widely used across various domains to explore how electromagnetic waves interact with different materials. Its applications span across photonic crystals, waveguides, plasmonics, and metamaterials. In our research, we harness the FDTD simulations for the design and optimization of optical devices, focusing on applications in solar cells, anti-reflection, and transparent electromagnetic interference shielding.

**Solar Cells.** Nanomaterials are revolutionizing solar cell technology, promising significantly enhanced efficiency and reduced costs. Certain nanostructures have the potential to surpass the Shockley-Queisser efficiency limit and capture light beyond the Yablonovitch or Lambertian limits [48, 59, 78, 9]. In this paper we investigate vertical nanowire arrays and nanosphere coatings, as they have proven abilities to enhance light trapping [32, 17, 72, 74, 19]. Our evaluation of solar cells is based on their ultimate efficiency, excluding transport and radiative losses, effectively focusing on optimizing solar absorption.

**Anti-Reflection.** Light traveling from air to glass partially reflects due to the disparity in index of refraction. Anti-reflection thin film coatings can be used to achieve perfect anti-reflection at a single wavelength and normal incidence angle. However, these structures are less effective for reflection across a range of wavelengths or incidence angles. This challenge has been tackled with moth eye [76, 63] and glasswing butterfly wing-inspired structures [60, 20], which contain sub-wavelength structures with an effective refractive index that gradually transits between those of the air and the glass. These structures achieve broad-spectrum and wide-angle anti-reflection, enhancing light emission from displays and improving light intake for photodetectors and solar cells.

**Electromagnetic Interference Shielding.** As the usage of electronic devices has grown, there is a growing demand for strategies to shield these devices from external electromagnetic waves and interference, which can lead to disruptions. Furthermore, transparent electromagnetic interference shielding has emerged as a significance field of interest in this context. However, a considerable challenge is balancing effective shielding performance and transparency [75, 40]. Notably, multi-layer films have been identified as a viable solution, successfully attaining high levels of both transparency and shielding effectiveness [42, 75]. Furthermore, the integration of nanocone structures into these films enhances their transparency, offering an innovative approach for this complex challenge [39].

## 2.3 Related Work

The field of molecular discovery has garnered extensive attention across numerous studies, leading to significant advancements [54]. One such contribution is the creation of a comprehensive dataset featuring a large array of organic molecules, complete with details on their geometric, energetic, electronic, and thermodynamic properties [52]. This dataset serves as a valuable resource for framing and addressing the challenges associated with molecular discovery. MoleculeNet, a tool developed using the DeepChem software, further extends this by consolidating various public molecular property datasets [77]. Moreover, benchmarks specifically designed for de novo molecular design have been introduced to aid in solving inverse molecular discovery problems [8].

The emergence of deep learning for nanophotonic device design, particularly in inverse design methods, has been discussed [62]. This era has also witnessed exploration into data-driven surrogate models for artificial electromagnetic materials, utilizing neural networks to capture the intricate behaviors of metamaterials, nanophotonics, and color filters [11]. A noteworthy study has conducted a comprehensive comparison of various graph neural networks, evaluating their effectiveness in learning the dynamics of simple physical systems [66]. Furthermore, a combination of Bayesian optimization and Bayesian neural networks has been effectively applied to tackle complex scientific challenges such as photonic crystal topology and quantum chemistry [34].

In the fields of machine learning and optimization, the community often leverages diverse benchmark functions. These include simple, cost-effective synthetic functions such as Branin, Rastrigin, and Ackley functions. The benchmarks extend beyond these elementary examples, encompassing a variety of scenarios for applications like black-box optimization [24, 68], hyperparameter optimization [14, 46, 57], and neural architecture search [80, 12, 36, 13]. Despite the diversity, a noticeable gap exists, as these benchmarks predominantly feature similar types of examples and problems, offering limited insight into the complex challenges prevalent in real-world scientific and engineering contexts.

## 3 Nanophotonic Structures and Their Design Problems

In this section we introduce the scheme of electrodynamic simulations, our nanophotonic structures, and their design problems. These include specifying possible material choices and ranges of sizes

for structure components. Moreover, we detail the consideration of simulation fidelity and multiple objectives. We also visualize how our objectives vary as we change parameters that define a structure.

## 3.1 Electrodynamic Simulations

We cover the scheme of electrodynamic simulations using the FDTD method – a widely adopted technique in time-domain simulations for nanophotonic applications. This method discretizes both spatial and temporal dimensions, enabling the approximation of derivatives at discrete points. Our simulations focus on exploring the optical behaviors of structures in two and three dimensions through the FDTD method [65].

For a light source, we employ a Gaussian-pulse wave, monitoring the resulting electric fields $\boldsymbol{E}(t)$ and magnetic fields $\boldsymbol{B}(t)$. These fields are subsequently Fourier-transformed to yield their frequency-dependent counterparts $\boldsymbol{E}(\omega)$ and $\boldsymbol{B}(\omega)$. A Yee grid, a computational mesh where the electric and magnetic field components are calculated, is used [79]. Our simulations are conducted using Meep, which is the open-source FDTD simulation software licensed under the GNU General Public License [44].

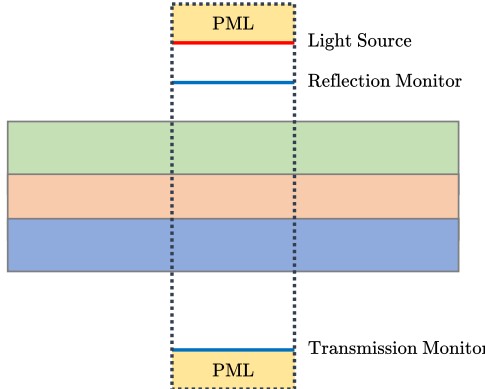

Figure 1: Schematic of nanophotonic structure simulations. PMLs, light source, and reflection and transmission monitors are located in a simulation cell indicated by the dotted rectangle.

Figure 1 shows a schematic that illustrates the setup of our FDTD simulation, where the simulation domain is confined within the simulation cell indicated by the dotted rectangle. The size and type of this simulation cell are tailored to the specific nanophotonic structure investigated. We utilize periodic boundary conditions for the sides of the simulation cell and perfectly matched layers (PMLs) for the top and bottom of the cell. Periodic boundary conditions model semi-infinite arrays, while the PML boundary conditions ensure that fields radiate to infinity instead of reflecting [5].

Within the simulation cell, a light source is positioned near the top, while transmission and reflection monitors are situated near the bottom and below the source, respectively. The electromagnetic wave is chosen to be normally incident to the structure of interest. The monitors enable the calculation of flux from $\mathrm{Re} \int \boldsymbol{E}^*(\omega) \times \boldsymbol{B}(\omega)/2$ as a function of frequency, facilitating the derivation of reflectance $R(\omega)$, absorbance $A(\omega)$, and transmittance $T(\omega)$. To accurately measure reflectance, the total incident flux from an empty simulation cell is subtracted from the flux measured in the presence of the nanophotonic structure. The positioning of the light source and monitors is chosen to ensure that they are appropriately spaced from the top and bottom boundaries of the simulation cell.

For our simulations, we consider three types of spectral ranges: a single wavelength 550 nm at the approximate center of the visible spectrum, the visible spectrum from 380 nm to 750 nm with standard illuminant D65 [27], and the AM1.5 global solar spectrum from 280 nm to 2500 nm [2].

## 3.2 Structures of Interest

In this section, we describe the specifics of the nanophotonic structures studied in this paper. The details of simulation cell sizes are presented in Section F. Table 1 provides the summary of the design space including parameters, materials, and parameters' ranges for each of these structures.

**Three-Layer Film for Electromagnetic Interference Shielding.** This film structure is aimed at mitigating electromagnetic interference, incorporating three distinct layers; see Figure 2a. The central layer is metallic, potentially consisting of silver (Ag), gold (Au), copper (Cu), or nickel (Ni). The adjacent layers can be selected from materials such as titanium dioxide ($TiO_2$), crystalline silicon (cSi), zinc oxide (ZnO), indium tin oxide (ITO), or aluminum-doped zinc oxide (AZO). We simplify our material selection by assuming the materials for the outer layers are identical. The three layers are characterized by their thicknesses: $t_1$, $t_2$, and $t_3$. We use the electromagnetic wave of wavelength $\lambda = 550$ nm, which approximately corresponds to green light.

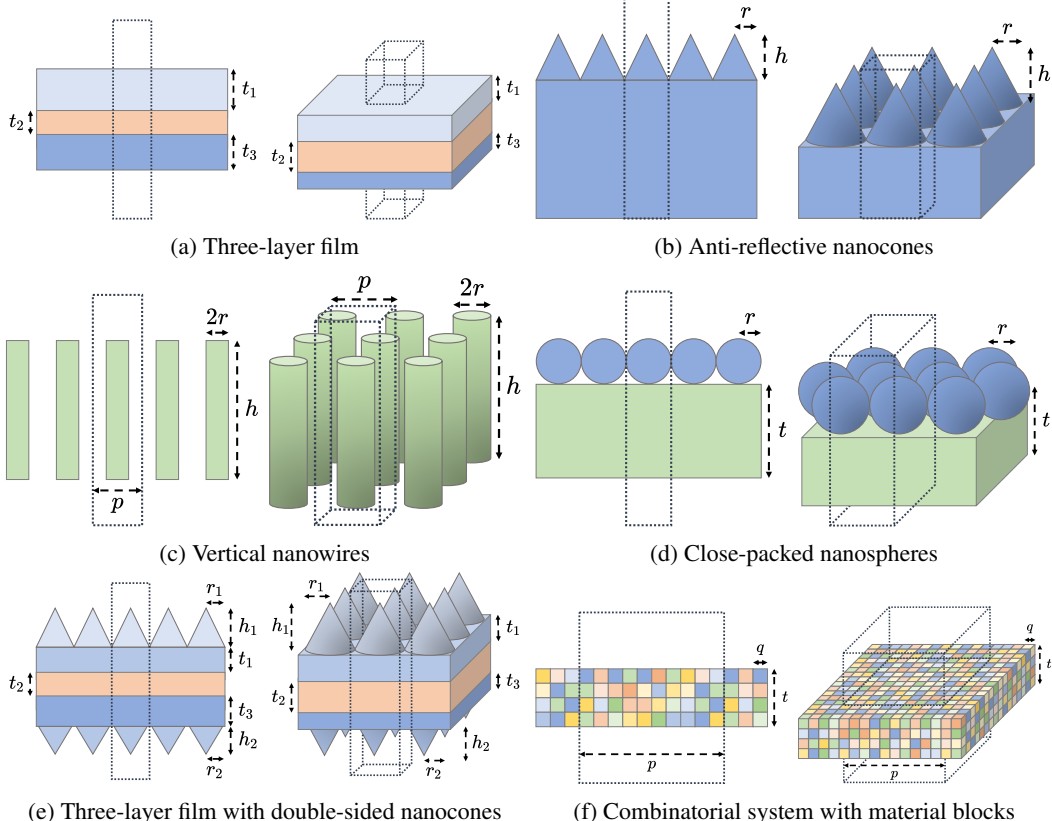

(a) Three-layer film

(b) Anti-reflective nanocones

(c) Vertical nanowires

(d) Close-packed nanospheres

(e) Three-layer film with double-sided nanocones

(f) Combinatorial system with material blocks

Figure 2: Two- and three-dimensional nanophotonic structures. Different colors indicate different materials in each figure and a dotted region represents a simulation cell.

The objectives of this system include maximizing transmittance over $t_1$, $t_2$, and $t_3$ and tackling a multi-objective optimization problem for maximizing both electromagnetic interference shielding effectiveness and transmittance. The formula for shielding effectiveness is $S = 20 \log_{10}(1 + \eta_0 t_2 / 2\rho)$, where $\eta_0 = 376.73 \ \Omega$ is the free space impedance and $\rho$ is the bulk metal resistivity.

**Graded Index of Refraction Structures for Anti-Reflection.** Here, we include a semi-infinite array of glass nanocones attached to a glass substrate, as illustrated in Figure 2b. The nanocones serve to gradually transition the index of refraction from air to glass, with fused silica as the glass of choice. This structure is defined by two parameters: the nanocones' height $h$ and radius $r$. Assuming a close-packed square array, we determine the pitch of the nanocones is $p = 2r$. Our goal is to minimize solar reflection; see Section C for the details of solar reflection.

**Vertical Nanowires for Solar Cells.** We explore arrays of vertical nanowires composed of crystalline silicon (cSi), gallium arsenide (GaAs), or a perovskite structure of methylammonium lead iodide ($CH_3NH_3PbI_3$), as shown in Figure 2c. The structures are defined by the nanowire radius $r$, height $h$, and pitch $p$, with a fixed $h = 200$ nm. The relationship $2r \leq p$ has to be satisfied, which makes the diameter less than or equal to the pitch. We redefine this constraint using $g = p - 2r$, where $g \geq 0$. The nanowires are arranged in a square array, with optimization focused on maximizing solar absorption or ultimate efficiency; refer to Section D for further details.

**Close-Packed Nanospheres for Solar Cells.** This system features titanium dioxide ($TiO_2$) nanospheres in a hexagonal array on top of a semiconductor thin film with possible materials including crystalline silicon (cSi), gallium arsenide (GaAs), or a perovskite structure of methylammonium lead iodide ($CH_3NH_3PbI_3$). The system, which is visualized in Figure 2d, is defined by the semiconductor layer thickness $t$ and nanosphere radius $r$. Similar to the vertical nanowires, the optimization aims at maximizing solar absorption or ultimate efficiency; refer to Section D for details.

Table 1: Parameters, materials, and the search spaces of the parameters for nanophotonic structures. We assume that only one of AZO, cSi, ITO, $TiO_2$, and ZnO should be used for one structural configuration in the three-layer film with or without double-sided nanocones.

| Structure | Parameter | Materials | Lower Bound (nm) | Upper Bound (nm) |
|---|---|---|---|---|
| Three-layer film | $t_1$ | AZO, cSi, ITO, $TiO_2$, ZnO | 10 | 100 |
| | $t_2$ | Ag, Au, Cu, Ni | 3 | 20 |
| | $t_3$ | AZO, cSi, ITO, $TiO_2$, ZnO | 10 | 100 |
| Anti-reflective nanocones | $r$ | Fused silica | 5 | 150 |
| | $h$ | | 1 | 300 |
| Vertical nanowires | $g$ | | 1 | 200 |
| | $r$ | cSi, $CH_3NH_3PbI_3$, GaAs | 5 | 200 |
| | $h$ | | 200 | 200 |
| Close-packed nanospheres | $t$ | cSi, $CH_3NH_3PbI_3$, GaAs | 100 | 400 |
| | $r$ | $TiO_2$ | 10 | 200 |
| Three-layer film with double-sided nanocones | $t_1$ | AZO, cSi, ITO, $TiO_2$, ZnO | 10 | 50 |
| | $t_2$ | Ag, Au, Cu, Ni | 3 | 20 |
| | $t_3$ | AZO, cSi, ITO, $TiO_2$, ZnO | 10 | 50 |
| | $r_1$ | AZO, cSi, ITO, $TiO_2$, ZnO | 20 | 50 |
| | $h_1$ | | 50 | 100 |
| | $r_2$ | AZO, cSi, ITO, $TiO_2$, ZnO | 20 | 50 |
| | $h_2$ | | 50 | 100 |
| Combinatorial system with material blocks | _ | Ag, Air, Au, AZO, cSi, $CH_3NH_3PbI_3$, Cu, GaAs, ITO, Ni, $TiO_2$, ZnO | _ | _ |

**Three-Layer Film with Nanocones for Electromagnetic Interference Shielding.** Building upon the three-layer film concept, this structure incorporates double-sided nanocones for enhanced electromagnetic interference shielding; see Figure 2e. It comprises three layers and nanocones on both sides, where material options for the three-layer film are identical to the previously discussed three-layer film and nanocones are made of one of titanium dioxide ($TiO_2$), crystalline silicon (cSi), zinc oxide (ZnO), indium tin oxide (ITO), and aluminum-doped zinc oxide (AZO). Seven parameters define this structure: three layer thicknesses $t_1$, $t_2$, $t_3$, and two sets of cone heights $h_1$, $h_2$ and radii $r_1$, $r_2$. Objective for the optimization of this structure can be either maximization of visible transmission or simultaneous maximization of visible transmission and shielding effectiveness, where light of the visible spectrum is injected; refer to the description of the aforementioned three-layer film.

**Combinatorial System with Material Blocks.** We introduce a novel and challenging problem: a combinatorial system composed of material blocks; see Figure 2f. The size of each block within the structure is $(q, q)$ for two-dimensional systems or $(q, q, q)$ for three-dimensional ones. Construction of this structure is accomplished by strategically selecting specific materials for all the blocks placed in a repeating unit. This unit consists of $p/q$ blocks along the $x$ and $y$ directions, and $t/q$ blocks along the $z$ direction, where $p$ denotes the repeating unit's pitch size and $t$ represents the maximum thickness of the structure. Consequently, the repeating unit is comprised of either $p/q \times t/q$ blocks for two-dimensional structures or $p/q \times p/q \times t/q$ blocks for three-dimensional ones.

In this paper, we choose specific values: $p = 200$ nm, $t = 40$ nm, and $q = 10$ nm. The range of materials considered for the material blocks includes silver (Ag), air, gold (Au), aluminum-doped zinc oxide (AZO), crystalline silicon (cSi), methylammonium lead iodide perovskite ($CH_3NH_3PbI_3$), copper (Cu), gallium arsenide (GaAs), indium tin oxide (ITO), nickel (Ni), titanium dioxide ($TiO_2$), and zinc oxide (ZnO). The AM1.5 solar spectrum is used for simulating this structure.

### 3.3 Simulation Fidelity and Multiple Objectives

As described above, the accuracy of simulation results are heavily influenced by the fidelity of the simulations. Simulations with a low fidelity level are faster but less accurate, while simulations with

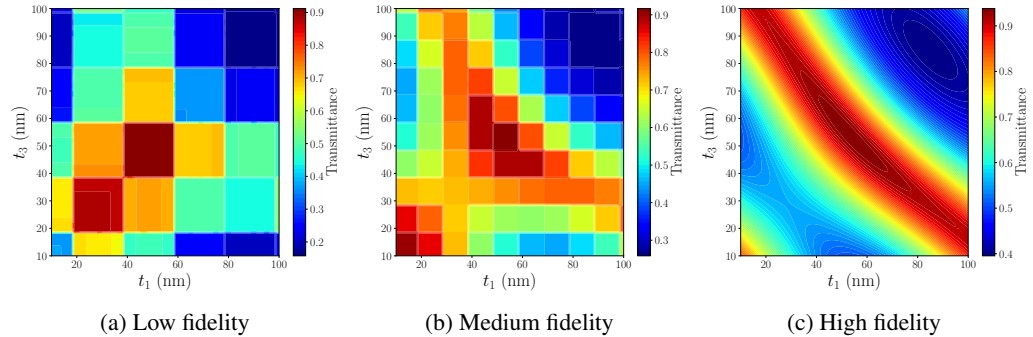

| (a) Low fidelity | (b) Medium fidelity | (c) High fidelity |

Figure 3: Visualization of the transmittance of the three-layer film made of $TiO_2/Ag/TiO_2$ for three different fidelity levels, where the second layer's thickness $t_2$ is 3 nm.

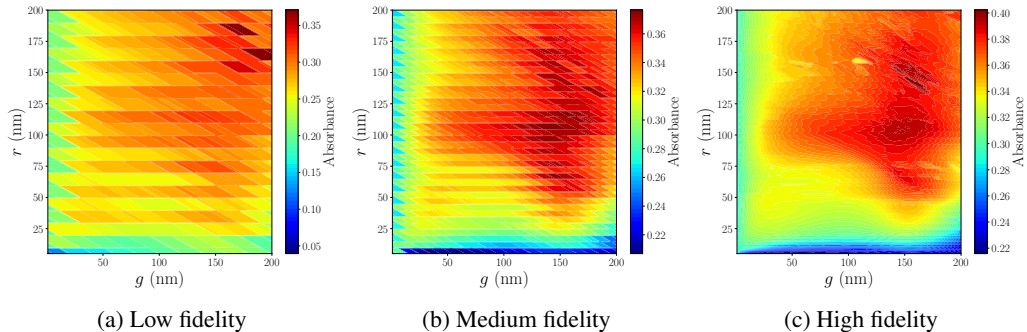

| (a) Low fidelity | (b) Medium fidelity | (c) High fidelity |

Figure 4: Visualization of the absorbance of the vertical nanowires made of cSi for three different fidelity levels, where the nanowire height $h$ is 200 nm.

a high fidelity level are more accurate but slower. There is an inherent tradeoff between speed and accuracy. Additionally, our frameworks can generate objective functions such as shielding effectiveness and transparency for three-layer films, both with and without nanocones. The introduction of multiple objectives adds a layer of complexity to the optimization problem. In summary, utilizing fidelity levels and multiple objectives enables us to expand our research, potentially exploring multi-fidelity optimization [15, 31, 4] and multi-objective optimization [37, 26, 3].

## 3.4 Visualization

For each type of nanophotonic structure featured in our frameworks, we visualize how specific optical properties change across different structural parameters, as shown in Figures 3, 4, and 6. It is important to note that the accuracy and smoothness of the optical property values are dependent on the fidelity level. Results from lower fidelity simulations tend to be less accurate and show more erratic fluctuations compared to results from their higher fidelity counterparts. This difference is evident in the comparisons provided in Figures 3 and 4. Additional visualization for more diverse examples is available in Section H, where the data and examples provided in this paper pertain to two-dimensional nanophotonic structures.

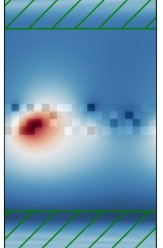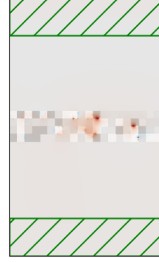

Figure 5: Examples of E-fields (left) and H-fields (right) out of plane for the combinatorial system with material blocks.

In addition to optical properties, our frameworks are also equipped to simulate and provide data on electric and magnetic fields (i.e., E- and H-fields), across various positions and time steps within a simulation cell. Figures 5 and 7 provide the visual representation of these fields, clearly showing how they are influenced and altered by the presence of nanophotonic structures.

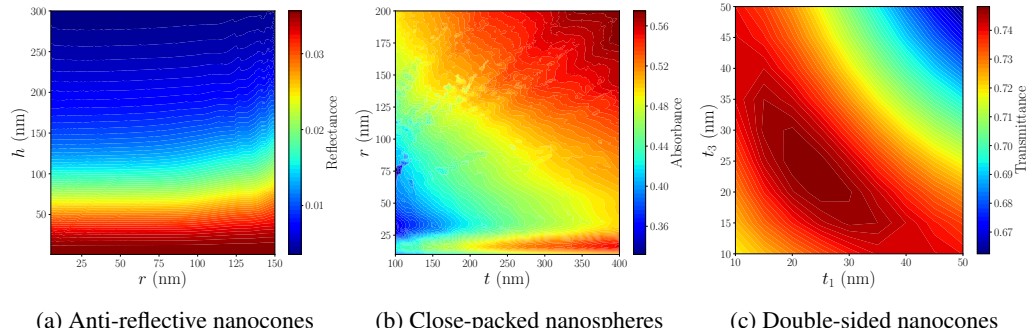

(a) Anti-reflective nanocones     (b) Close-packed nanospheres     (c) Double-sided nanocones

Figure 6: Visualization of the target properties of anti-reflective nanocones, close-packed nanospheres, and three-layer film with nanocones, made of fused silica, cSi/$TiO_2$, and $TiO_2$/Ag/$TiO_2$/$TiO_2$/$TiO_2$, respectively. For the film with nanocones, $t_2 = 3$ nm, $r_1 = r_2 = 20$ nm, $h_1 = h_2 = 50$ nm.

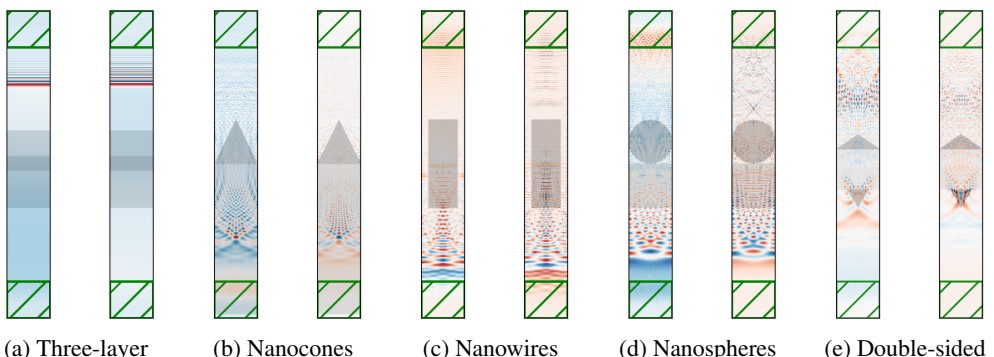

(a) Three-layer    (b) Nanocones    (c) Nanowires    (d) Nanospheres    (e) Double-sided

Figure 7: Examples of E-fields (left of each panel) and H-fields (right of each panel) out of plane for the structures studied in this work. The results for the combinatorial system are shown in Figure 5.

## 4   Datasets and Benchmarks for Nanophotonic Structures and Their Designs

Following Section 3, we present our datasets and benchmarks defined with several aforementioned nanophotonic structures and their design problems in this section.

### 4.1   Datasets

Our frameworks provide datasets and processes for generating datasets, tailored for nanophotonic structures and their associated parametric design simulations. To generate these datasets, it is necessary to define feasible search spaces for parametric structure designs. Building on Table 1, Table 2 presents the number of parameters associated with each structure, the increment value for each structure, and the number of possible configurations conducted for each structure.

Our frameworks provide datasets of two primary modalities:

- Reflection, absorption, and transmission spectra, accompanied by fidelity information: This includes measurements of the three key properties across various structural configurations, considering fidelity information. The fidelity level, which can be low, medium, or high, is determined by the simulation resolution as defined by the spacing of the Yee grid.

- Electric and magnetic fields: These E- and H-fields are measured across different positions and simulation time steps as a light source emits a Gaussian-pulse light until it decays sufficiently. The resolution of the simulation and the size of the simulation cell determine the number of grid points used for these positional and temporal measurements.

By leveraging the diverse components within our datasets, users can explore and test a variety of tasks, ranging from multi-fidelity and multi-objective optimization to direct property predictions. In

Table 2: Details of our discretization of the search spaces for the datasets and discretized search space optimization mode. Elaborate description on the number of configurations can be found in Section G.

| Structure | #Parameters | Increment (nm) | #Configurations |
|---|---|---|---|
| Three-layer film | 3 | 1 | 149,058 |
| Anti-reflective nanocones | 2 | 1 | 43,800 |
| Vertical nanowires | 3 | 1 | 39,200 |
| Close-packed nanospheres | 2 | 1 | 57,491 |
| Three-layer film with double-sided nanocones | 7 | 5 | 1,920,996 |
| Combinatorial system with material blocks | – | – | $12^{80}$ or $12^{1600}$ |

particular, our datasets can serve as a valuable testbed for evaluating a variety of machine learning models, such as training them to predict optical spectra or electromagnetic fields, based on material properties and geometric configurations.

To provide readily accessible datasets for practitioners, we select some material combinations from Table 1 and then generate datasets. Specifically, for the three-layer films with and without double-sided nanocones, we assume that the top and bottom layers, and potentially the top and bottom nanocones are made of the identical material. Due to the vast number of possible configurations for the combinatorial system with material blocks, we do not create datasets for this structure, but instead provide a specific feature to work with this structure; see Section 4.2 for further details.

## 4.2 Optimization Modes

Our benchmarks support two modes for fast prototyping of optimization methods as follows.

**Discretized Search Space Mode.** In this mode, a search space is discretized with the increment specified in Table 2. At each configuration a simulation is run and the outcome of the simulation is recorded. Access to these simulation results is available online.

**Surrogate Model Mode.** Based on the dataset collected from the discretized search space, we fit a surrogate model. This allows us to evaluate any structural configuration in the search space using the trained surrogate model. All the specifics of this process are detailed in Section J. It is crucial to note that the predictions made by the surrogate model are not actual evaluations and might contain errors. A comprehensive discussion of these limitations and their implications can be found in Section N.

Note that we currently support these two modes only for the selection of some material combinations. Also, we do not support these modes for the combinatorial system with material blocks because the search space of this system is extremely huge and therefore it is difficult to obtain its datasets and build an accurate surrogate model.

**Simulation Mode.** Users can also directly run these simulations using our frameworks based on Meep and perform an optimization algorithm by conducting a simulation at every iteration. This can be more time consuming to execute but is preferred if the user has sufficient compute available. This allows users to carry out an optimization procedure on the complete space specified in Table 1 including exploring different material choices.

## 4.3 Experiments with the Surrogate Model Mode

In this section, we evaluate the performance of various optimization methods applied to parametric structure optimization. We consider a range of approaches including random search, Powell's method [49], Py-BOBYQA [10], DIRECT [30], differential evolution [64], and Bayesian optimization [16]. For the experiments shown in Figure 8, we specifically focus on assessing the effectiveness of our benchmarks in the surrogate model mode that is defined on a continuous search space. Most of the algorithms are implemented using SciPy [69] and NumPy [25] and some algorithms are executed using the open-source versions of them; see Section K for their missing details. Each experiment is independently repeated 50 times, with consistent random seeds maintained across all methods. The detailed description of experimental setup and implementation can be found in Sections J and K.

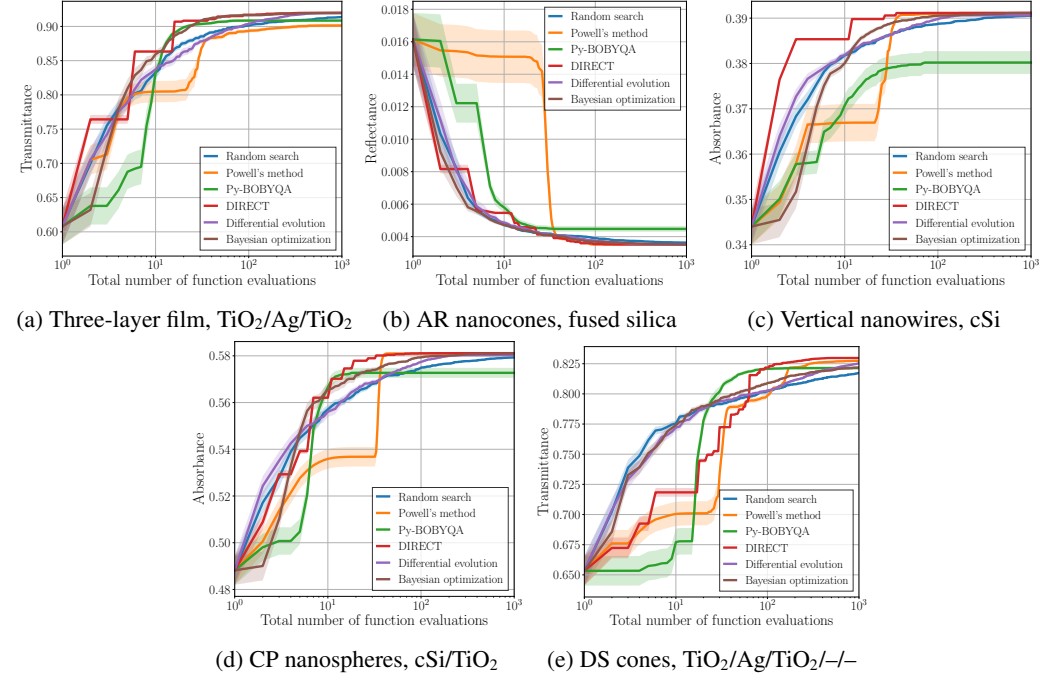

(a) Three-layer film, $TiO_2/Ag/TiO_2$   (b) AR nanocones, fused silica   (c) Vertical nanowires, cSi

(d) CP nanospheres, $cSi/TiO_2$   (e) DS cones, $TiO_2/Ag/TiO_2/–/–$

Figure 8: Results of experiments on nanophotonic structure optimization. Each experiment is conducted 50 times and the mean and standard error are depicted. AR nanocones, CP nanospheres, DS cones indicate the anti-reflective nanocones, close-packed nanospheres, and three-layer film with double-sided nanocones, respectively. Also, $TiO_2/Ag/TiO_2/–/–$ stands for $TiO_2/Ag/TiO_2/TiO_2/TiO_2$.

The experimental results presented in Figure 8 show that global optimization methods such as DIRECT, differential evolution, and Bayesian optimization generally perform well across different structures. DIRECT, in particular, consistently outperforms the other techniques. Bayesian optimization and differential evolution also exhibit strong performance, although they can be slower than or comparable to DIRECT. However, in the case of the three-layer film with nanocones, we find that DIRECT tends to be slower than both Bayesian optimization and differential evolution; see Figure 8e. Conversely, local optimization methods such as Powell's method and Py-BOBYQA generally take longer to converge and are prone to getting stuck in local optima. While Powell's method converges to the same solutions as DIRECT in some cases such as Figures 8b, 8c, and 8d, both local search methods struggle to escape local optima in the other instances. This is the expected behavior of local optimization methods. Interestingly, random search, despite its simplicity, demonstrates effectiveness and even outperforms the local optimization methods in particular cases. However, its performance drops in the higher-dimensional problem; see Figure 8e.

On the other hand, for the combinatorial system with material blocks, we conduct experiments with random search using the simulation mode to evaluate configurations directly. The results and the analysis on these results are described in Section L.

## 5   Conclusion

In this paper we introduced several nanophotonic structures such as the three-layer film, anti-reflective nanocones, vertical nanowires, close-packed nanospheres, three-layer film with double-sided nanocones, and combinatorial system with material blocks for real-world applications. To construct our datasets and benchmarks, we devised a generic simulation scheme and pipeline for nanophotonic structure and parametric design simulations. Finally, the datasets and benchmarks are proposed by modeling, simulating, and optimizing nanophotonic structures. Furthermore, the future directions, limitations, and societal impacts of our work are discussed in Sections M, N, and O.

## Acknowledgments and Disclosure of Funding

This research was partly funded by the National Science Foundation (NSF) under grant ECCS 1552712. We also received support from the MDS-Rely Center, which is funded by the NSF's Industry–University Cooperative Research Center (IUCRC) program through awards EEC-2052662 and EEC-2052776. OH acknowledges support from the National Science Foundation and United States-Israel Binational Science Foundation (NSF-BSF) program under NSF grant 2239527 and from Air Force Office of Scientific Research (AFOSR) grant FA9550-23-1-0242. Additionally, the University of Pittsburgh Center for Research Computing provided essential resources for this work, particularly the H2P cluster. This cluster operates with support from the NSF award OAC-2117681.

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

## A Electromagnetic Wave of Light

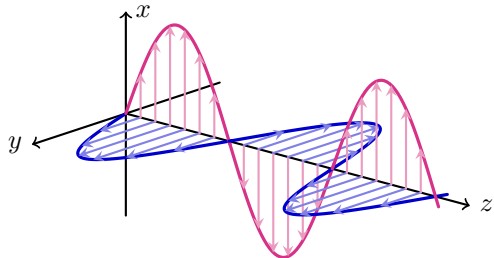

Figure S1: Electromagnetic wave of light. Magenta and blue waves correspond to electric and magnetic fields, respectively.

Figure S1 shows the electromagnetic wave of light, which is composed of electric and magnetic fields. The oscillations of these two fields are perpendicular to each other and the direction of wave propagation.

## B Reflection, Absorption, and Transmission of Light

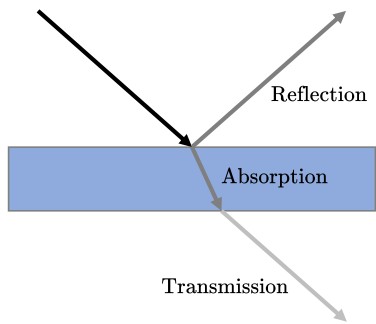

Figure S2: Light interacting with some material.

Figure S2 depicts the properties of light, where light interacts with some material. As in (2), the sum of reflectance, absorbance, and transmittance is one.

## C Solar Reflection

Solar reflection $R_{\text{solar}}$ over $d$ parameters $x_1, x_2, \ldots, x_d$ is defined as the following:

$$R_{\text{solar}}(x_1, x_2, \ldots, x_d) = \frac{\int b_s(\lambda) R(x_1, x_2, \ldots, x_d; \lambda) \, \mathrm{d}\lambda}{\int b_s(\lambda) \, \mathrm{d}\lambda}, \tag{S1}$$

where $R(x_1, x_2, \ldots, x_d; \lambda)$ is a reflection function of wavelength $\lambda$ and $b_s(\lambda)$ is the photon flux density of the AM1.5 spectrum [2] at wavelength $\lambda$.

## D Solar Absorption and Ultimate Efficiency

Solar absorption $A_{\text{solar}}$ and ultimate efficiency $\eta_{\text{ue}}$ over $d$ parameters $x_1, x_2, \ldots, x_d$ are defined as the following:

$$A_{\text{solar}}(x_1, x_2, \ldots, x_d) = \frac{\int_{E_g}^{\infty} b_s(E) A(x_1, x_2, \ldots, x_d; E) \, \mathrm{d}E}{\int_0^{\infty} b_s(E) \, \mathrm{d}E}, \tag{S2}$$

$$\eta_{\text{ue}}(x_1, x_2, \ldots, x_d) = \frac{\int_{E_g}^{\infty} I(E) A(x_1, x_2, \ldots, x_d; E) \frac{E_g}{E} \, \mathrm{d}E}{\int_0^{\infty} I(E) \, \mathrm{d}E}, \tag{S3}$$

where $b_s(E)$ is a photon flux density, $I(E)$ is the irradiance, and $E_g$ is the materials' energy band gap. Note that ultimate efficiency assumes that the temperature of the solar cell is 0 K, and $\int_0^\infty I(E)\, dE = 1{,}000$ W/m$^2$ for the AM1.5 global spectrum [2]. Each material can only absorb light with energy above its band gap $E_g$ so the integration in the numerator is only for photons with energy greater than $E_g$. $E_g = 1.12$ eV ($\lambda = 1107$ nm), for crystalline silicon (cSi), $E_g = 1.43$ eV ($\lambda = 867$ nm) for gallium arsenide (GaAs), and $E_g = 1.51$ eV ($\lambda = 821$ nm) for a perovskite structure of methylammonium lead iodide (CH$_3$NH$_3$PbI$_3$).

Consequently, maximizing $A_{\text{solar}}$ is equivalent to maximizing $\eta_{\text{ue}}$:

$$\max A_{\text{solar}}(x_1, x_2, \ldots, x_d) = \max \eta_{\text{ue}}(x_1, x_2, \ldots, x_d). \tag{S4}$$

## E  Material Properties

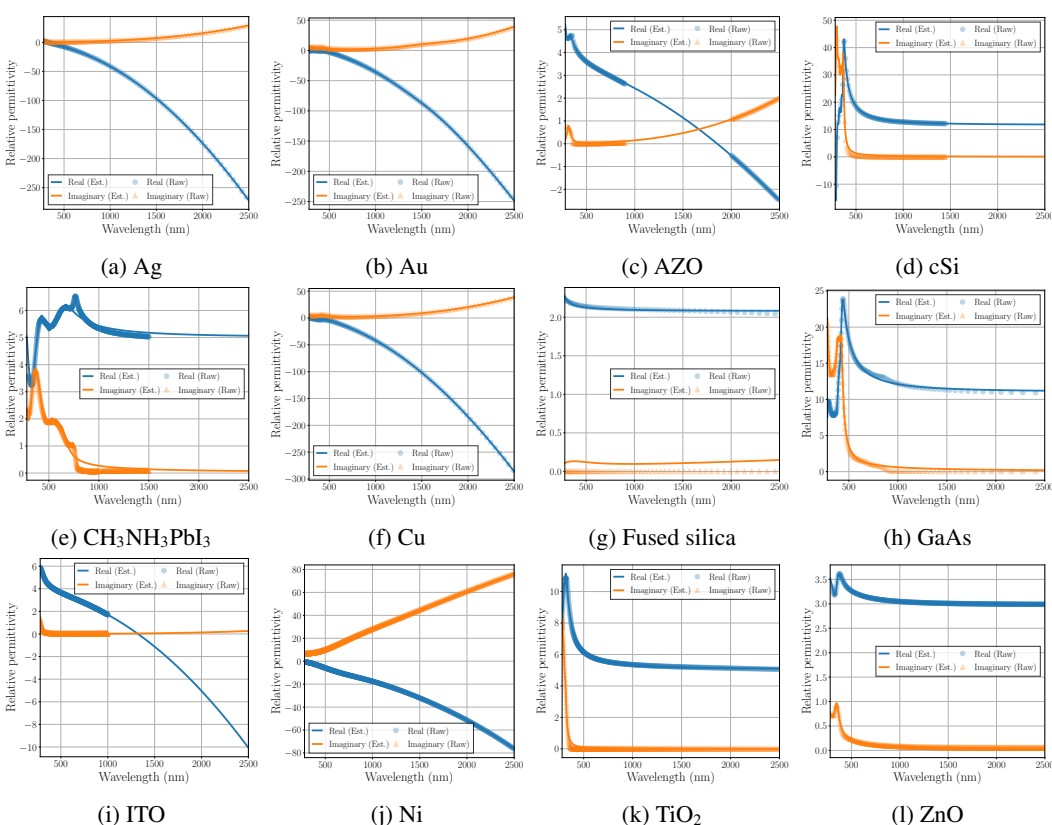

Figure S3: Relative permittivities of various materials versus different wavelengths. We utilize the raw values of complex permittivity and employ a gradient-based optimization technique to fit them to the Drude-Lorentz susceptibility model. The estimated complex permittivities presented here cover the solar spectrum range from 280 nm to 2500 nm.

In Figure S3, we report relative permittivities across different wavelengths for the materials employed in this work. The range of the solar spectrum we focus on spans from 280 nm to 2500 nm. To derive permittivity values throughout this range, we apply a gradient-based optimization method to fit the raw complex permittivity data to the Drude-Lorentz susceptibility model, following the methodology outlined in the Meep documentation [44].

For AZO, we combine two different sets of raw permittivity data to create a comprehensive dataset. This approach aims to enhance the accuracy of the fitted model across the entire solar spectrum. The effectiveness of this approach and the results of the fitted model are illustrated in Figure S3c.

With regard to fused silica, a minor offset is intentionally introduced to the optical extinction coefficient of its complex refractive index. This adjustment is necessary to stabilize the simulations,

especially those at low and medium fidelity levels, which are otherwise susceptible to divergence or failure. In this research, we choose an offset value of 0.04, and the associated results and adjustments are detailed in Figure S3g.

The materials examined in this paper include silver (Ag) [51], gold (Au) [51], aluminum-doped zinc oxide (AZO) [67, 58], crystalline silicon (cSi) [55], a perovskite structure of methylammonium lead iodide ($CH_3NH_3PbI_3$) [47], copper (Cu) [51], fused silica [41], gallium arsenide (GaAs) [50], indium tin oxide (ITO) [38], nickel (Ni) [51], titanium dioxide ($TiO_2$) [61], and zinc oxide (ZnO) [1]. Note that the respective references of the materials mentioned above indicate the sources of the raw data of the relative permittivities.

## F    Details of Simulation Cell Sizes

Table S1: Details of the simulation cell sizes declared for our simulations. A unit depth $m$ determines the spacing of geometries. All values are in nanometers. For simplicity, denote that $t^\dagger = t_2 + \max(2t_1, 2t_3) + \max(2h_1, 2h_2)$.

| Structure | Two-Dimensional Structure | Three-Dimensional Structure |
|---|---|---|
| Three-layer film | $(10, t_2 + \max(2t_1, 2t_3) + 6m)$ | $(10, 10, t_2 + \max(2t_1, 2t_3) + 6m)$ |
| Anti-reflective nanocones | $(2r, 2h + 6m)$ | $(2r, 2r, 2h + 6m)$ |
| Vertical nanowires | $(p, h + 6m)$ | $(p, p, h + 6m)$ |
| Close-packed nanospheres | $(2r, \max(4r, 2t) + 6m)$ | $(2r, 2r\sqrt{3}, \max(4r, 2t) + 6m)$ |
| Three-layer film with nanocones | $(\max(2r_1, 2r_2), t^\dagger + 6m)$ | $(\max(2r_1, 2r_2), \max(2r_1, 2r_2), t^\dagger + 6m)$ |
| Combinatorial system | $(p, t + 6m)$ | $(p, p, t + 6m)$ |

Table S1 represents the sizes of the simulation cells employed for nanophotonic structures. A unit depth $m$ determines spacing between geometries such as the light source, transmission monitor, reflection monitor, and photonic structures. More precisely, supposing that we define the depth of PML as $m$, distance between the light source and the reflection monitor is set as $m$ and the size of the simulation cell is also dependent on $m$; see Table S1 for that dependency. Unless otherwise noted, $m = 50$ nm for our simulations.

## G    Details of Datasets

The number of possible configurations shown in Table 2 is determined by considering the search space presented in Table 1 and increment for the corresponding structure. For example, suppose that we are given two parameters where the first parameter can vary between 3 and 5, and the second between 10 and 14, increasing in steps of 1. In this example, the first parameter has three possible values {3, 4, 5}, and the second has five values {10, 11, 12, 13, 14}. This results in a total of 15 unique configurations that can be derived from these parameters.

## H    Additional Dataset Visualization

Along with the visualizations presented in Section 3.4, additional examples pertaining to various nanophotonic structures are provided in this section. For the three-layer film structure, we display simulation results with different material compositions in Figures S5 through S12. Specifically, these illustrate variations in the nanophotonic structure when different materials are used in its composition. Next, we present simulation results for the anti-reflective nanocones in Figure S13, showcasing variations in this particular structure type. This is followed by simulation results for the vertical nanowires, exhibited in Figure S14, and the close-packed nanospheres, shown in Figure S15. Furthermore, we provide the visualization of simulations results for the three-layer film with double-sided nanocones in Figures S16 through S23. Similar to other visualization, these demonstrate how the results change with different material compositions. It is worth noting that for nanophotonic structures characterized by more than two parameters, we vary two of the parameters while keeping the others constant. This approach allows us to effectively represent their parameter spaces in two-dimensional plots.

# I  Results on Elapsed Time

Table S2: Results on elapsed time for diverse nanophotonic structures and fidelity levels. Mean and standard deviation are reported. All values are in seconds.

| Structure | Low Fidelity | Medium Fidelity | High Fidelity |
|---|---|---|---|
| Three-layer film, $TiO_2$/Ag/$TiO_2$ | $0.5927 \pm 0.0810$ | $1.1277 \pm 0.1373$ | $54.4757 \pm 23.1198$ |
| Anti-reflective nanocones, fused silica | $0.5999 \pm 0.1357$ | $0.6762 \pm 0.1583$ | $60.6909 \pm 54.9886$ |
| Vertical nanowires, cSi | $1.0668 \pm 0.2669$ | $1.5037 \pm 0.4536$ | $377.0777 \pm 215.7626$ |
| Close-packed nanospheres, cSi/$TiO_2$ | $1.5695 \pm 3.2443$ | $2.3399 \pm 4.8280$ | $673.3744 \pm 848.0346$ |
| Film with nanocones, $TiO_2$/Ag/$TiO_2$/$TiO_2$/$TiO_2$ | $2.2194 \pm 0.7021$ | $2.3046 \pm 0.7497$ | $115.0109 \pm 57.6097$ |

We demonstrate results on elapsed time for the nanophotonic structures we study and three fidelity levels in Table S2. Since we run a large number of simulations through a job scheduler, i.e., the Slurm workload manager, the respective simulations can be run on distinct machines; see Section K. Although we assign the same number of threads for each job, the time elapsed for simulations might vary depending on the status of machines. Nevertheless, assuming that all simulation jobs are fairly distributed to the machines and machine states are also similar on average, the results on elapsed time are provided for analysis. As expected, simulations with low fidelity is faster than ones with medium fidelity and high fidelity, and simulations with high fidelity is slower than the others. Moreover, the time differences between low fidelity and medium fidelity are marginal compared to the differences between medium fidelity and high fidelity or between low fidelity and high fidelity.

# J  Details of Regression Models

To establish a regression model for predicting a target optical property, we begin by partitioning our dataset into subsets designated for training, validation, and testing. Specifically, these disjoint subsets are 70%, 10%, and 20% of the entire dataset, respectively.

The regression model for the surrogate model mode is a multi-layer perceptron, detailed as follows:

> First layer: (the number of parameters, 128, batch normalization, ReLU);
>
> Second layer: (128, 64, batch normalization, ReLU);
>
> Third layer: (64, 1, –, Logistic),

where $(x, y, f, g)$ implies (the number of input dimensions, the number of output dimensions, a normalization technique, an activation function). Training of this network is performed using the Adam optimizer [35] and the mean squared error, configured with a learning rate of 0.001 and a batch size of 64. The network architecture and its associated components including batch normalization [28] are implemented using PyTorch [45]. To enhance the training process and select an optimal model, we employ early stopping based on the validation dataset's performance. We set the maximum number of training epochs as 200 and use the average of the last five validation losses as a threshold to determine when to stop training early. This approach helps us to finalize our model by ensuring that it generalizes well without overfitting.

Upon completion of the training process, the final model's performance on the test dataset is evaluated in terms of mean squared errors, yielding the following results for various nanophotonic structures:

- Three-layer film made of $TiO_2$/Ag/$TiO_2$: mean squared error of 0.000549;
- Anti-reflective nanocones made of fused silica: mean squared error of 0.000004;
- Vertical nanowires made of cSi: mean squared error of 0.000058;
- Close-packed nanospheres made of cSi/$TiO_2$: mean squared error of 0.000061;
- Three-layer film with double-sided nanocones made of $TiO_2$/Ag/$TiO_2$/$TiO_2$/$TiO_2$: mean squared error of 0.000020.

These results demonstrate the models' ability to reliably estimate the target optical properties across different structural configurations.

## K  Details of Experiments

In the main text, we delineate the comparative analysis of various optimization techniques applied to parametric structure optimization. The optimization algorithms used in this work can be enumerated as follows:

- Random search: a technique of selecting points from a uniform distribution;
- Powell's method [49]: a method to find a local solution without an assumption on differentiability;
- Py-BOBYQA [10]: a derivative-free approach for local optimization;
- DIRECT [30]: a global optimization method without the Lipschitz constant;
- Differential evolution [64]: a metaheuristic approach that iteratively improves solution candidates;
- Bayesian optimization [16]: a probabilistic model-based global optimization strategy for black-box functions.

For the implementation of random search, we employ the NumPy's uniform distributions [25], where NumPy is under the BSD license. Powell's method, DIRECT, and differential evolution are executed via their respective implementations in SciPy [69], also BSD licensed. Py-BOBYQA is run using the implementation obtained from its public repository,[S1] which operates under the GNU General Public License. For Bayesian optimization, we utilize BayesO [33], which is available under the MIT license. A variety of commercial CPUs, e.g., AMD EPYC 9374F, AMD EPYC 7302, and Intel Xeon Gold 6126, are used as the computational resources deployed for dataset creation, benchmarking, and structure optimization.

## L  Experiments for the Combinatorial System with Material Blocks

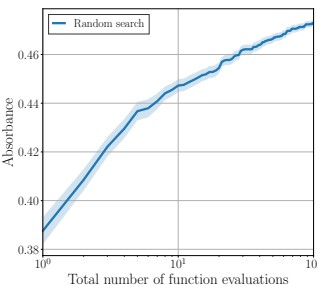

Figure S4: Results of structure optimization for the combinatorial system with material blocks using random search. Similar to Figure 8, it repeats each experiment 50 times. The sample mean and the standard error of the sample mean are plotted.

For the combinatorial system with material blocks, we test random search with uniform distributions in the process of selecting materials for all the material blocks in the structure. The objective of this problem is absorbance maximization for the AM1.5 global solar spectrum. To simplify the task, we dismiss band gaps for particular materials for these experiments. Since we have twelve options for materials, the number of possible structural configurations, i.e., in this case $12^{80}$, is enormous. This benchmark can be utilized to compare various combinatorial optimization algorithms, and the development and utilization of more advanced combinatorial optimization techniques are left for future work.

## M  Future Directions

Topology optimization via automatic differentiation has been studied in the field of photonics [22, 56]. Currently, we do not include support for such gradient-based methods. However, the FDTD

---

[S1]Accessible here: https://github.com/numericalalgorithmsgroup/pybobyqa.

simulation software including Meep [44] and Ansys Lumerical has added features for efficient gradient computation (with roughly twice the cost of a single function evaluation) [23]. Incorporating this into our benchmarks is a natural topic for future work.

In addition, we implement a user-friendly process for the integration of additional structures, supporting the involvement of a broad background of researchers and practitioners. We aim to constantly develop and maintain this work, and will strive to actively encourage the contributions of other researchers and practitioners.

## N    Limitations

The simulations we conduct are subject to potential instabilities and require precise settings. As a result, there are occasions when the combined values of reflectance, absorbance, and transmittance might exceed 1. It is important to note, however, that the results presented in this work do not yield any negative values in the reflectance, absorbance, and transmittance measurements we are concerned with. That said, it is within the realm of possibility for simulations to produce negative results.

Additionally, in contexts of continuous optimization with a surrogate model, the regression models we employ are not entirely devoid of errors, which can stem from both incorrect assumptions made by the model, i.e., inductive bias, and the inherent inaccuracies of the model itself. Eventually, these errors could introduce inaccuracies into the optimization process. Despite this potential for discrepancy between the surrogate model and the actual scenario, the comparisons we make between different optimization algorithms remain valid, provided that the surrogate model used is consistently applied across all optimization procedures.

## O    Societal Impacts

This research contributes to the field of materials science, specifically focusing on nanophotonic structures and their design problems. As such, it is important to consider the potential societal impacts, both negative and positive, that may arise from the practical application of our findings. On one hand, the manufacturing, fabrication, and disposal of materials and nanophotonic structures could lead to environmental pollution, affecting air, water, and land quality. It is crucial that these pollution issues are addressed proactively and managed effectively to mitigate any adverse effects on the environment and society. On the other hand, our work has the potential to significantly benefit the renewable energy sector, particularly in the advancement of solar cell technology. By contributing to the development of more efficient solar cells, we are aiding the transition towards a more sustainable energy landscape, which could result in a reduction of greenhouse gas emissions and a move towards a carbon-neutral society.

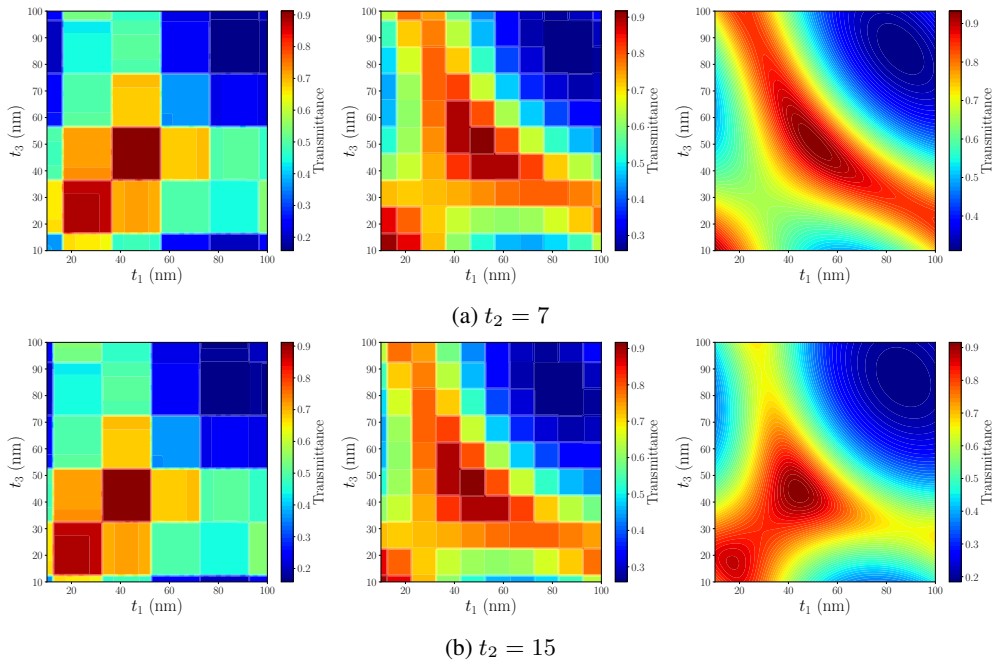

(a) $t_2 = 7$

(b) $t_2 = 15$

Figure S5: Visualization of the transmittance of the three-layer film made of TiO$_2$/Ag/TiO$_2$ for three different fidelity levels, i.e., low fidelity (shown in left panels), medium fidelity (shown in center panels), and high fidelity (shown in right panels).

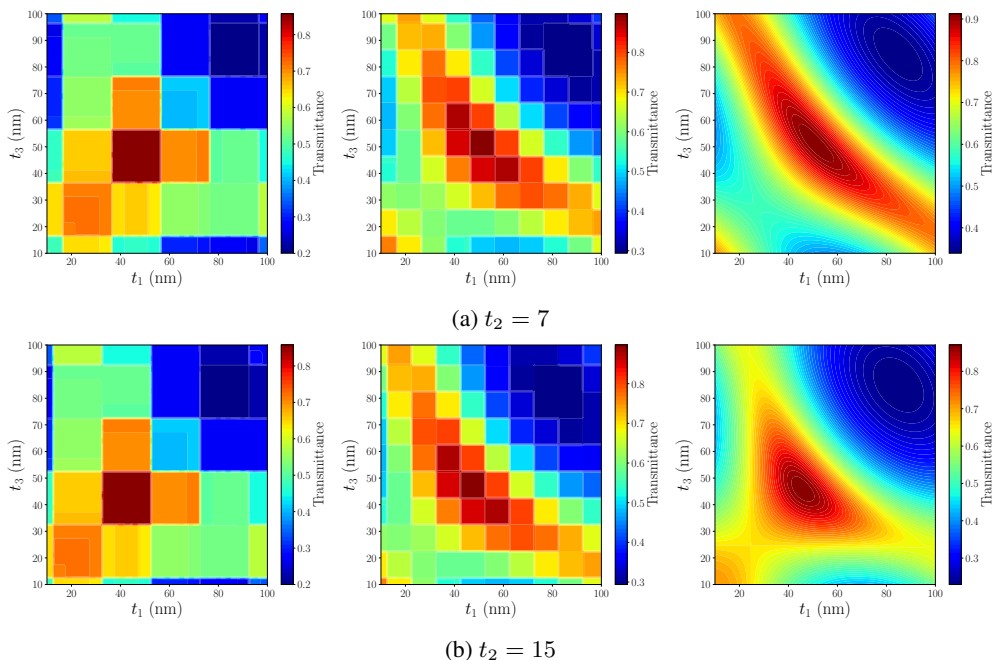

(a) $t_2 = 7$

(b) $t_2 = 15$

Figure S6: Visualization of the transmittance of the three-layer film made of TiO$_2$/Au/TiO$_2$ for three different fidelity levels, i.e., low fidelity (shown in left panels), medium fidelity (shown in center panels), and high fidelity (shown in right panels).

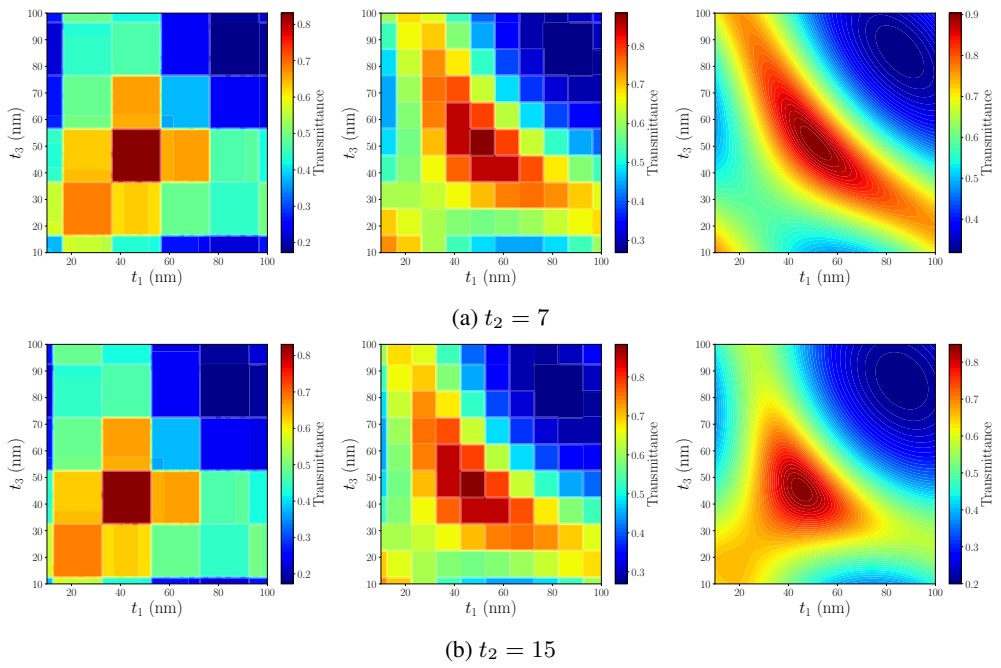

(a) $t_2 = 7$

(b) $t_2 = 15$

Figure S7: Visualization of the transmittance of the three-layer film made of $TiO_2/Cu/TiO_2$ for three different fidelity levels, i.e., low fidelity (shown in left panels), medium fidelity (shown in center panels), and high fidelity (shown in right panels).

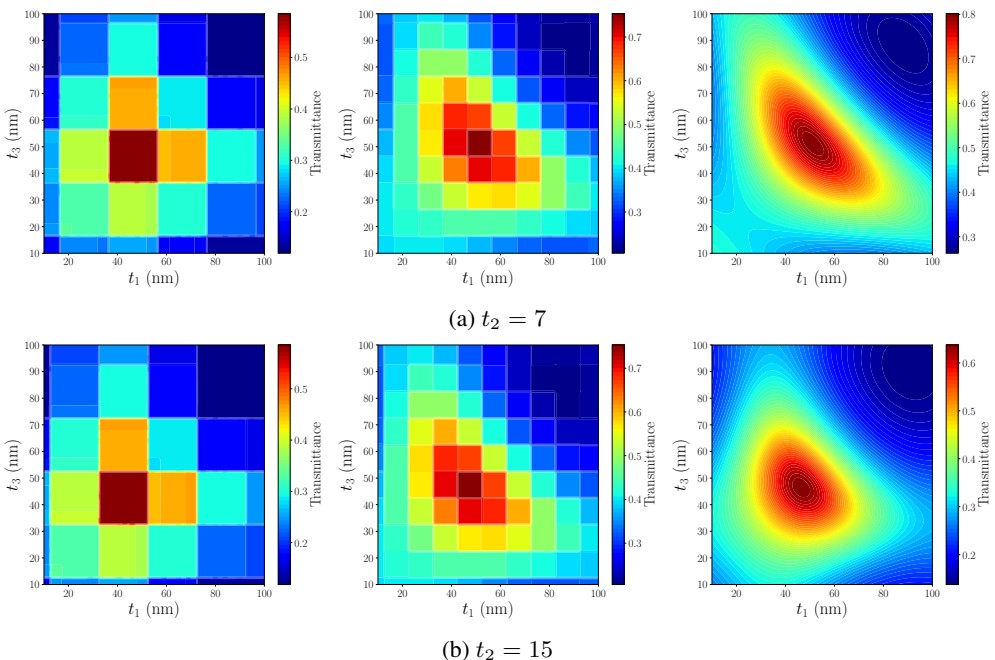

(a) $t_2 = 7$

(b) $t_2 = 15$

Figure S8: Visualization of the transmittance of the three-layer film made of $TiO_2/Ni/TiO_2$ for three different fidelity levels, i.e., low fidelity (shown in left panels), medium fidelity (shown in center panels), and high fidelity (shown in right panels).

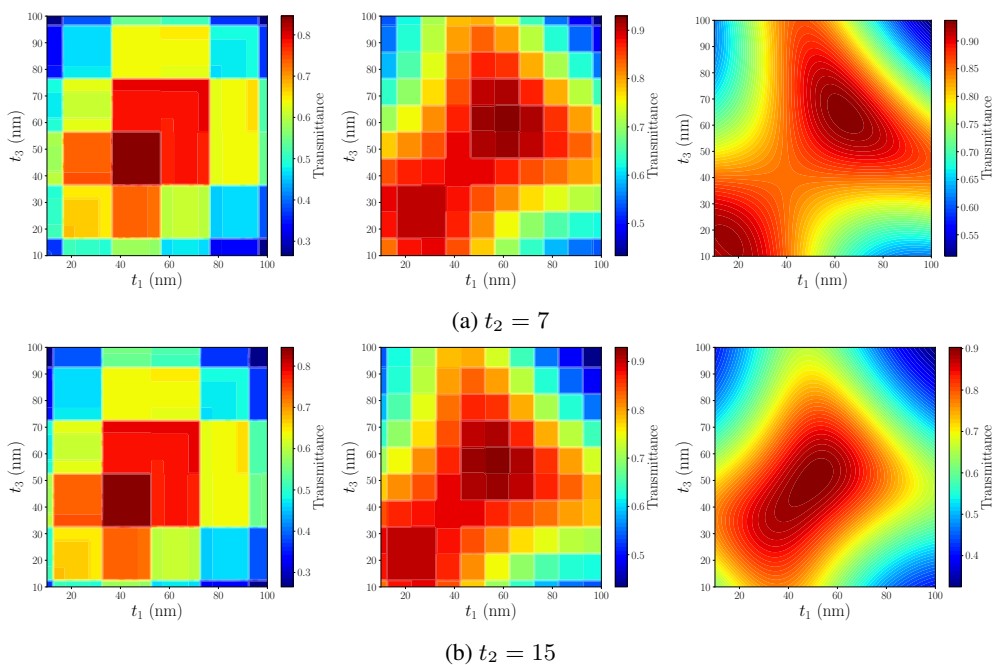

(a) $t_2 = 7$

(b) $t_2 = 15$

Figure S9: Visualization of the transmittance of the three-layer film made of AZO/Ag/AZO for three different fidelity levels, i.e., low fidelity (shown in left panels), medium fidelity (shown in center panels), and high fidelity (shown in right panels).

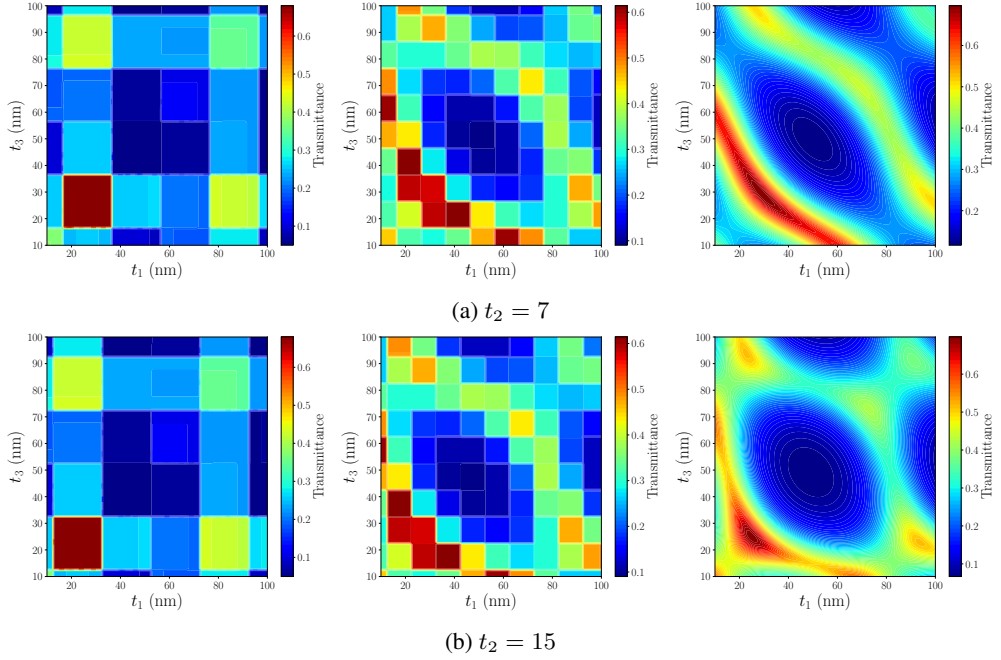

(a) $t_2 = 7$

(b) $t_2 = 15$

Figure S10: Visualization of the transmittance of the three-layer film made of cSi/Ag/cSi for three different fidelity levels, i.e., low fidelity (shown in left panels), medium fidelity (shown in center panels), and high fidelity (shown in right panels).

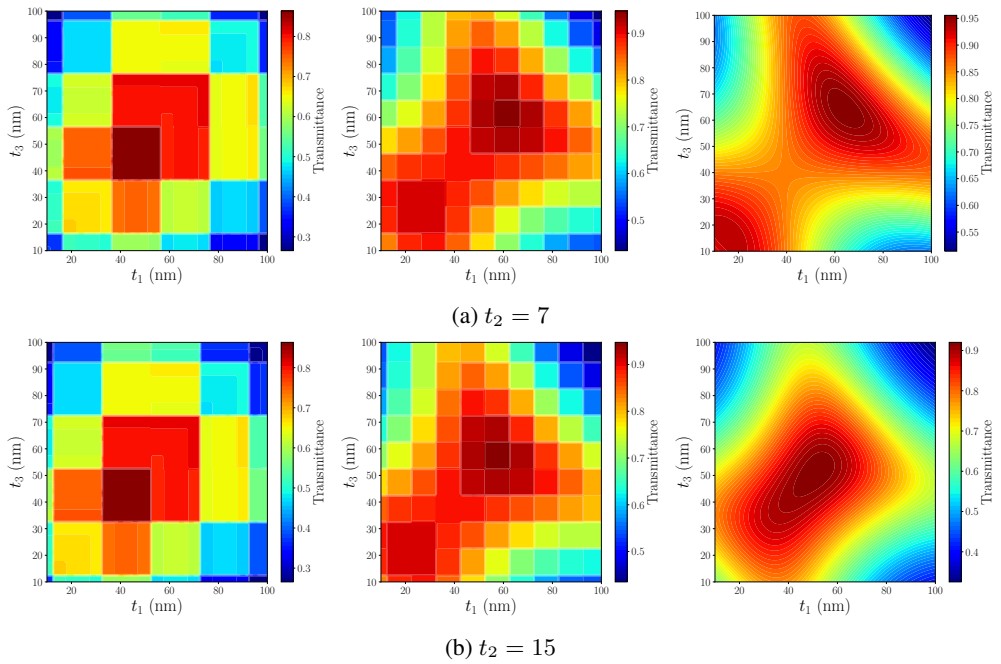

(a) $t_2 = 7$

(b) $t_2 = 15$

Figure S11: Visualization of the transmittance of the three-layer film made of ITO/Ag/ITO for three different fidelity levels, i.e., low fidelity (shown in left panels), medium fidelity (shown in center panels), and high fidelity (shown in right panels).

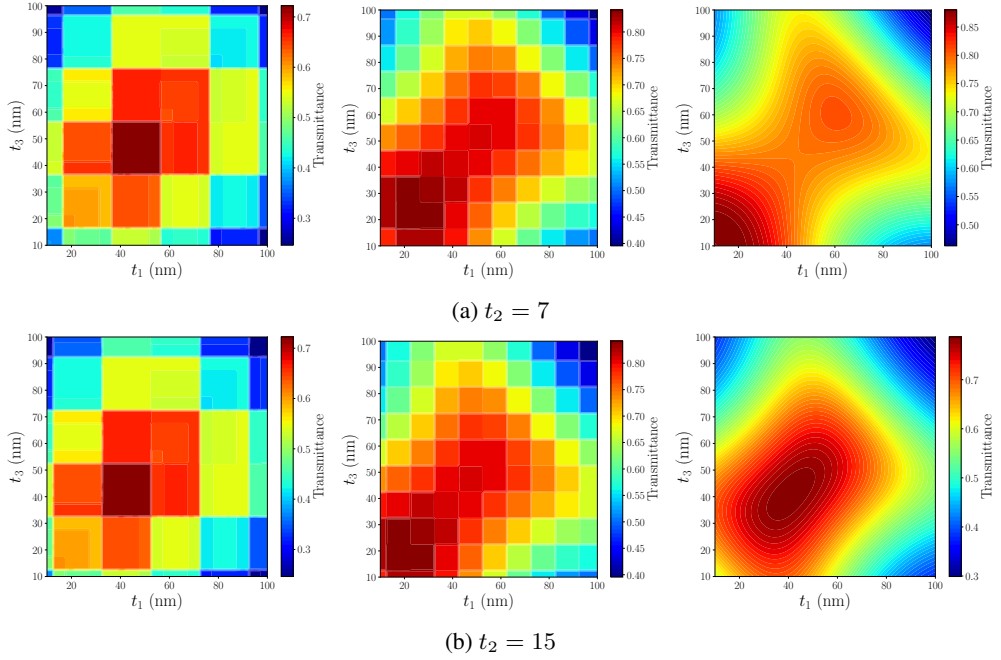

(a) $t_2 = 7$

(b) $t_2 = 15$

Figure S12: Visualization of the transmittance of the three-layer film made of ZnO/Ag/ZnO for three different fidelity levels, i.e., low fidelity (shown in left panels), medium fidelity (shown in center panels), and high fidelity (shown in right panels).

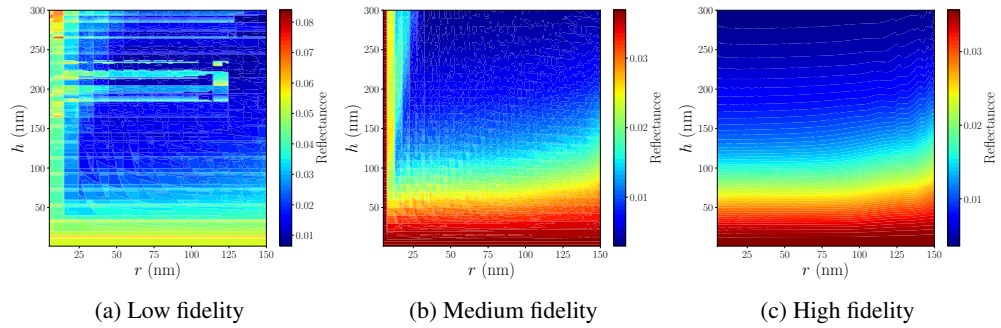

(a) Low fidelity        (b) Medium fidelity        (c) High fidelity

Figure S13: Visualization of the reflectance of the anti-reflective nanocones made of fused silica for three different fidelity levels.

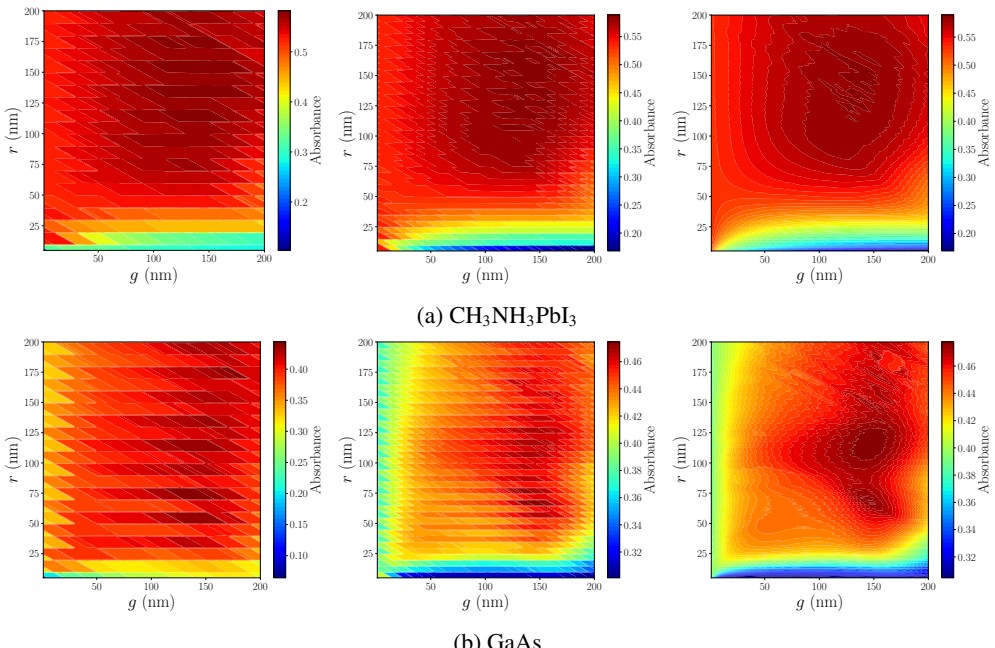

(a) $CH_3NH_3PbI_3$

(b) GaAs

Figure S14: Visualization of the absorbance of the vertical nanowires made of $CH_3NH_3PbI_3$ or GaAs for three different fidelity levels, i.e., low fidelity (shown in left panels), medium fidelity (shown in center panels), and high fidelity (shown in right panels).

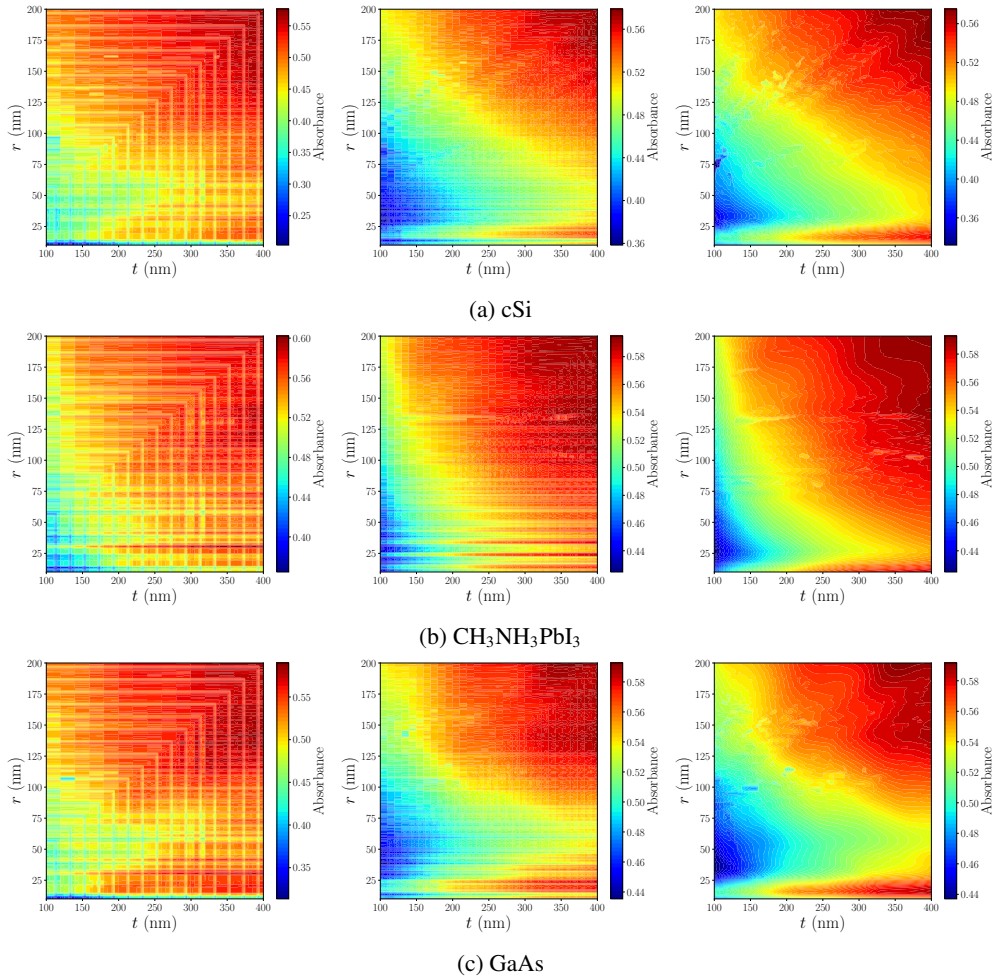

Figure S15: Visualization of the absorbance of the close-packed TiO$_2$ nanospheres on top of a thin film made of cSi, CH$_3$NH$_3$PbI$_3$, or GaAs for three different fidelity levels, i.e., low fidelity (shown in left panels), medium fidelity (shown in center panels), and high fidelity (shown in right panels).

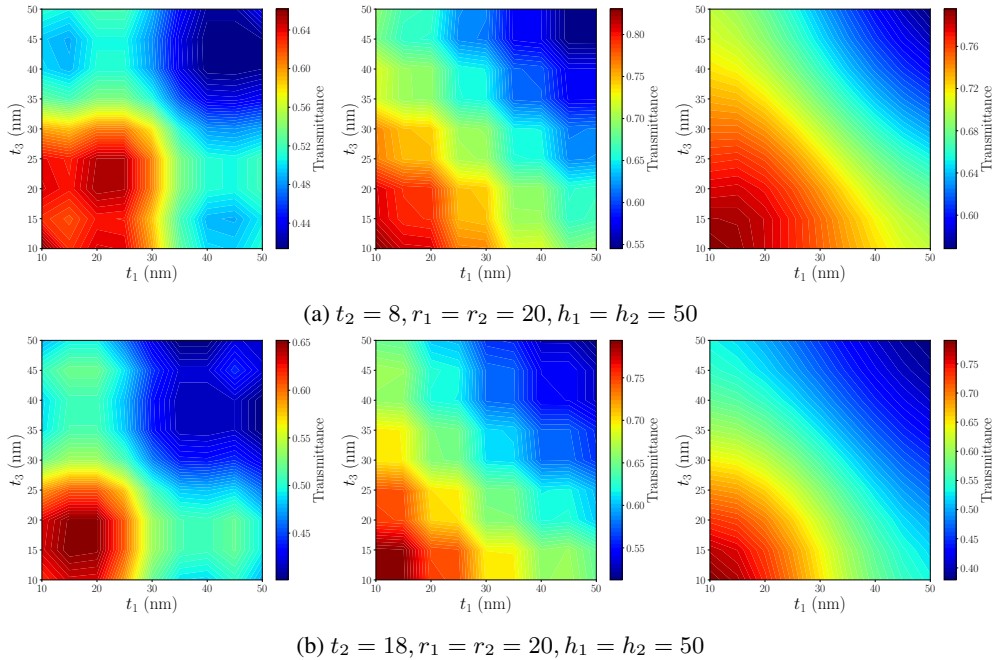

(a) $t_2 = 8, r_1 = r_2 = 20, h_1 = h_2 = 50$

(b) $t_2 = 18, r_1 = r_2 = 20, h_1 = h_2 = 50$

Figure S16: Visualization of the transmittance of the three-layer film with double-sided nanocones made of $TiO_2/Ag/TiO_2/TiO_2/TiO_2$ for three different fidelity levels, i.e., low fidelity (shown in left panels), medium fidelity (shown in center panels), and high fidelity (shown in right panels).

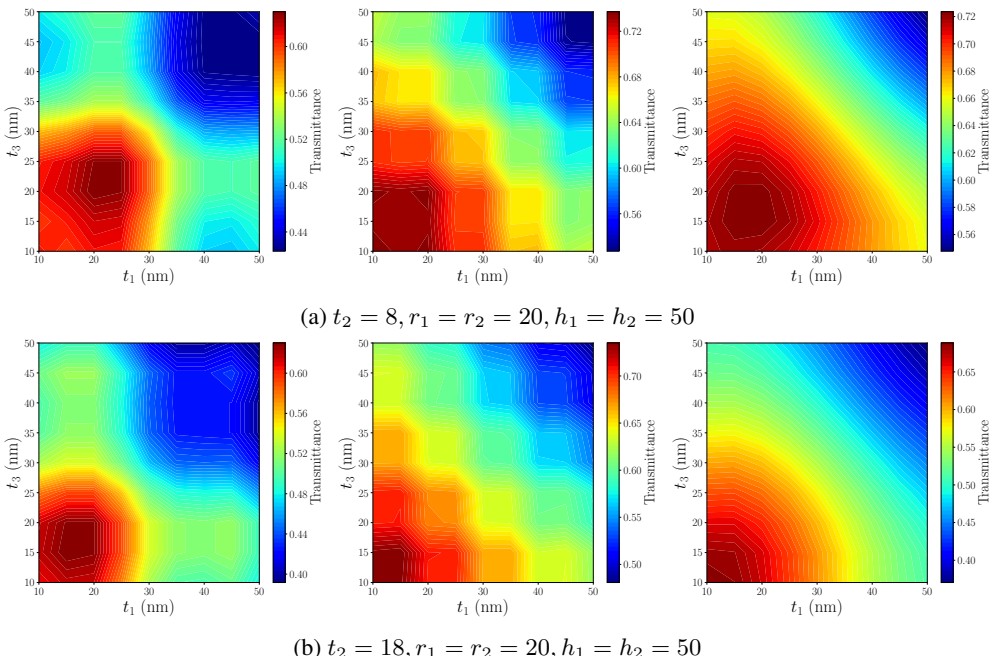

(a) $t_2 = 8, r_1 = r_2 = 20, h_1 = h_2 = 50$

(b) $t_2 = 18, r_1 = r_2 = 20, h_1 = h_2 = 50$

Figure S17: Visualization of the transmittance of the three-layer film with double-sided nanocones made of $TiO_2/Au/TiO_2/TiO_2/TiO_2$ for three different fidelity levels, i.e., low fidelity (shown in left panels), medium fidelity (shown in center panels), and high fidelity (shown in right panels).

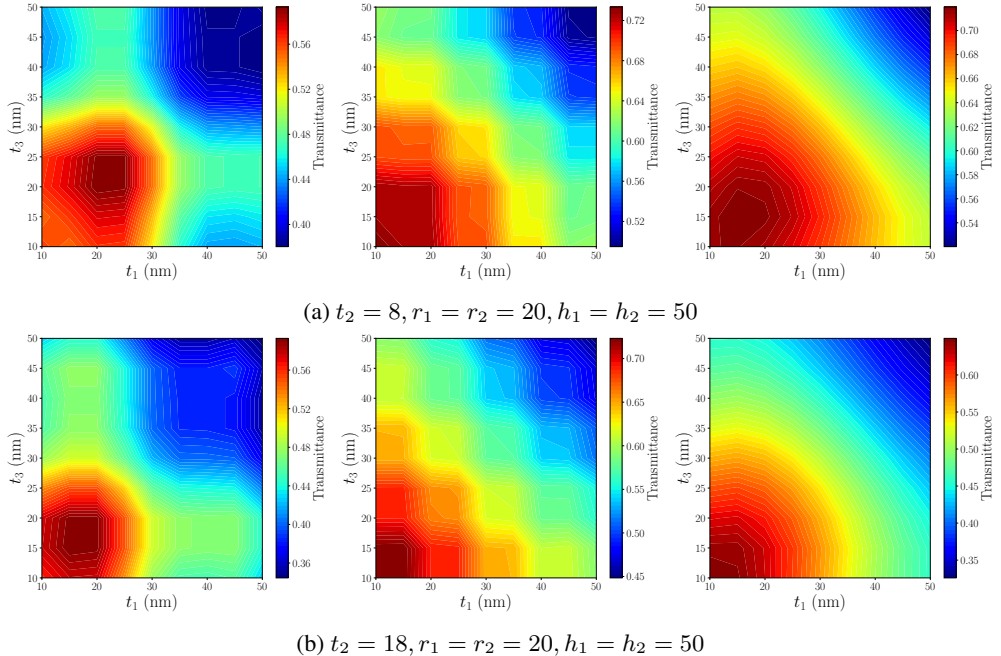

(a) $t_2 = 8, r_1 = r_2 = 20, h_1 = h_2 = 50$

(b) $t_2 = 18, r_1 = r_2 = 20, h_1 = h_2 = 50$

Figure S18: Visualization of the transmittance of the three-layer film with double-sided nanocones made of $TiO_2/Cu/TiO_2/TiO_2/TiO_2$ for three different fidelity levels, i.e., low fidelity (shown in left panels), medium fidelity (shown in center panels), and high fidelity (shown in right panels).

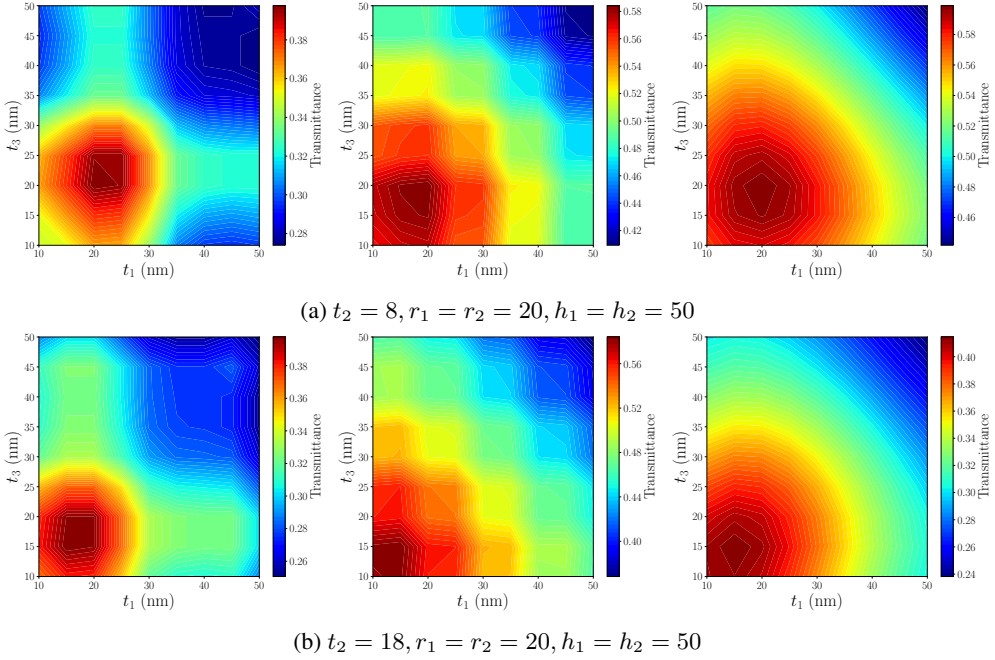

(a) $t_2 = 8, r_1 = r_2 = 20, h_1 = h_2 = 50$

(b) $t_2 = 18, r_1 = r_2 = 20, h_1 = h_2 = 50$

Figure S19: Visualization of the transmittance of the three-layer film with double-sided nanocones made of $TiO_2/Ni/TiO_2/TiO_2/TiO_2$ for three different fidelity levels, i.e., low fidelity (shown in left panels), medium fidelity (shown in center panels), and high fidelity (shown in right panels).

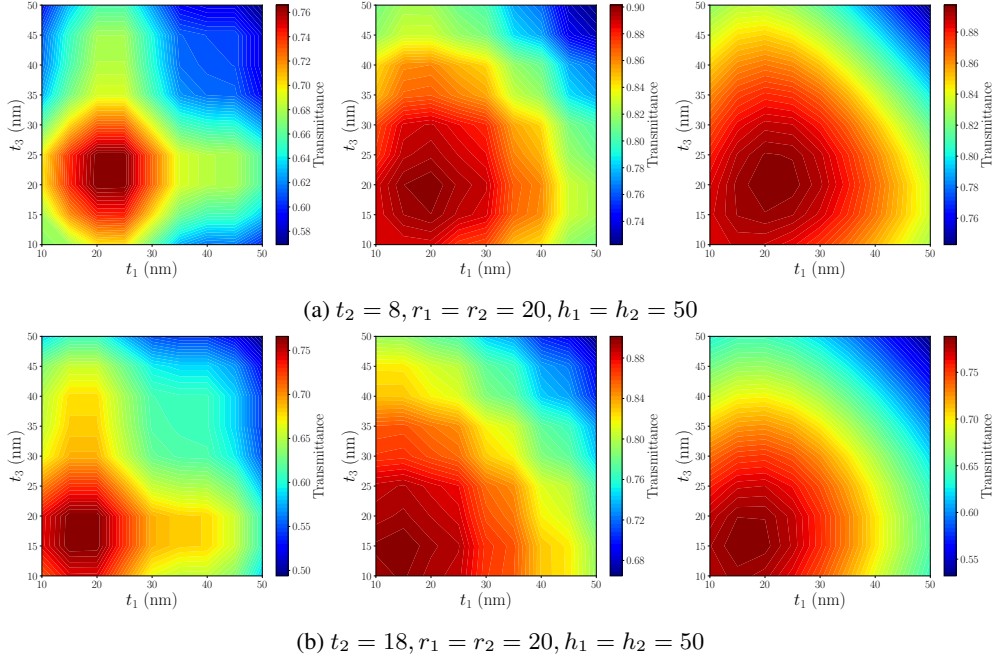

(a) $t_2 = 8, r_1 = r_2 = 20, h_1 = h_2 = 50$

(b) $t_2 = 18, r_1 = r_2 = 20, h_1 = h_2 = 50$

Figure S20: Visualization of the transmittance of the three-layer film with double-sided nanocones made of AZO/Ag/AZO/AZO/AZO for three different fidelity levels, i.e., low fidelity (shown in left panels), medium fidelity (shown in center panels), and high fidelity (shown in right panels).

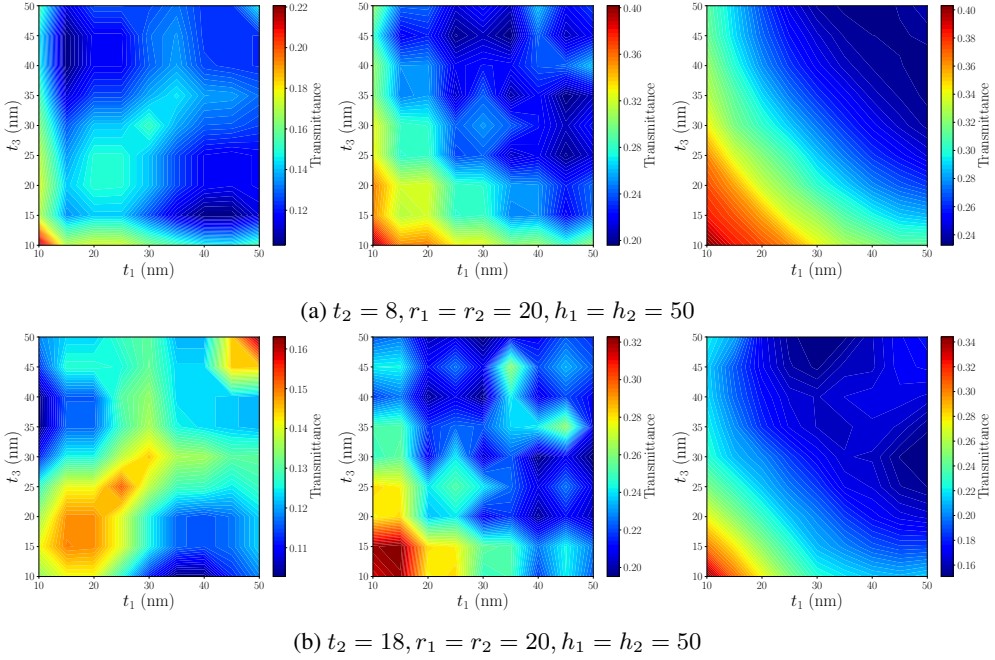

(a) $t_2 = 8, r_1 = r_2 = 20, h_1 = h_2 = 50$

(b) $t_2 = 18, r_1 = r_2 = 20, h_1 = h_2 = 50$

Figure S21: Visualization of the transmittance of the three-layer film with double-sided nanocones made of cSi/Ag/cSi/cSi/cSi for three different fidelity levels, i.e., low fidelity (shown in left panels), medium fidelity (shown in center panels), and high fidelity (shown in right panels).

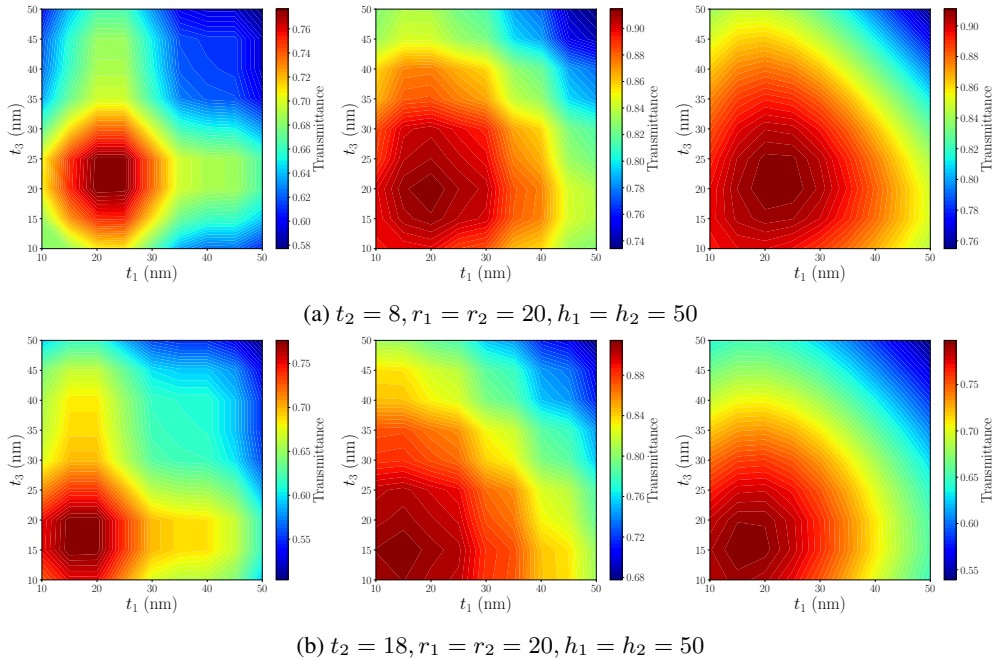

(a) $t_2 = 8, r_1 = r_2 = 20, h_1 = h_2 = 50$

(b) $t_2 = 18, r_1 = r_2 = 20, h_1 = h_2 = 50$

Figure S22: Visualization of the transmittance of the three-layer film with double-sided nanocones made of ITO/Ag/ITO/ITO/ITO for three different fidelity levels, i.e., low fidelity (shown in left panels), medium fidelity (shown in center panels), and high fidelity (shown in right panels).

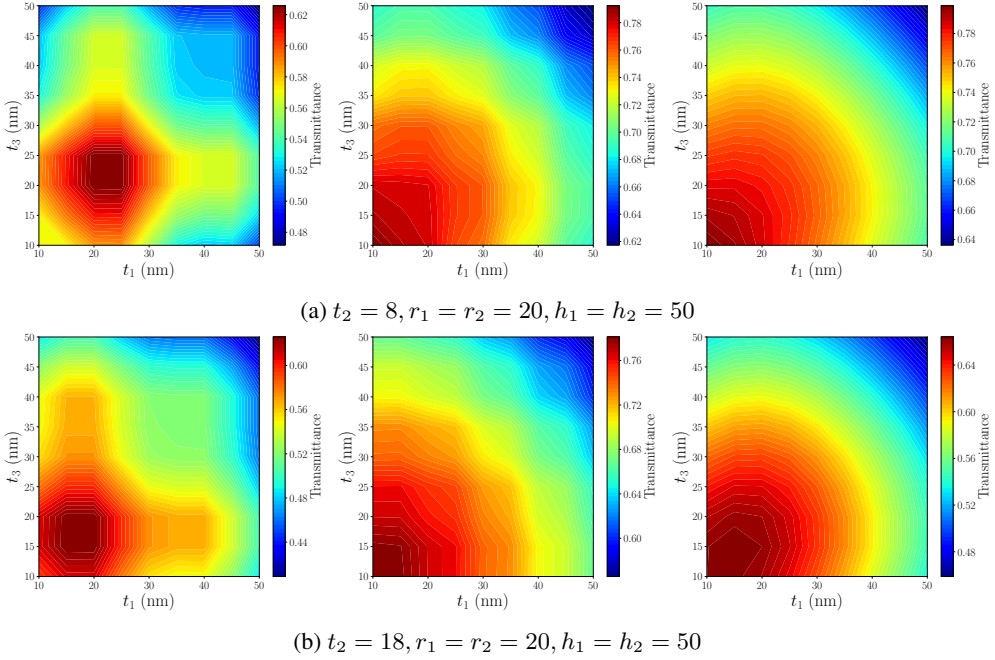

(a) $t_2 = 8, r_1 = r_2 = 20, h_1 = h_2 = 50$

(b) $t_2 = 18, r_1 = r_2 = 20, h_1 = h_2 = 50$

Figure S23: Visualization of the transmittance of the three-layer film with double-sided nanocones made of ZnO/Ag/ZnO/ZnO/ZnO for three different fidelity levels, i.e., low fidelity (shown in left panels), medium fidelity (shown in center panels), and high fidelity (shown in right panels).

