# OpenReview forum: "Datasets and Benchmarks for Nanophotonic Structure and Parametric Design Simulations"
_NeurIPS.cc/2023/Track/Datasets_and_Benchmarks — NeurIPS 2023 Datasets and Benchmarks Poster_

### Official Review · Reviewer_ZH3g · 2023-07-22
**Review of "Datasets and Benchmarks for Nanophotonic Structure and Parametric Design Simulations"**

**Rating:** 7
**Confidence:** 1
**Correctness:** The methodology and presented dataset…
**Clarity:** The paper is well written.

**Strengths:**

The paper is responsive to the conference topic, it is well written, the approach is sound, and overall methodology is appropriate. This submission merits acceptance.

**Additional Feedback:**

Paper is recommended for acceptance.

**Documentation:**

Sufficient description is presented.

**Ethics:**

No ethics concerns are noted.

**Limitations:**

Limitations are adequately addressed.

**Opportunities For Improvement:**

The paper can be accepted in the present form.

**Relation To Prior Work:**

Relation to prior work is sufficiently described.

**Summary And Contributions:**

This paper presents a benchmark and methodology for calculating the properties and design of photonic structures. The novel contributions of this paper are clear, the paper is responsive to the conference topic, it is well written, the approach is sound, and overall methodology is appropriate. This submission merits acceptance.

---

> ### Author Response · Authors · 2023-08-15
> **Response to Reviewer ZH3g**
>
> We thank the reviewer for your valuable comments.
>
> > Paper is recommended for acceptance.
>
> Thank you for your positive comment.  We uploaded the revision, the updated supplementary material, and the updated implementation.  Please take a look into them.

---

> > ### Author Response · Authors · 2023-08-23
> >
> > We thank you for your constructive comments.
> >
> > We revised our submission by considering your comments and the other reviews.
> >
> > If you have further concerns, please let us know.

---

### Official Review · Reviewer_wCnZ · 2023-07-24
**Datasets and Benchmarks for Nanophotonic Structure and Parametric Design Simulations**

**Rating:** 7
**Confidence:** 1
**Clarity:** Generally, yes.

**Strengths:**

The datasets and benchmarks appear to be useful to develop lower fidelity models targeted for the design of nano materials to form desired nanophotonic structures.

**Additional Feedback:**

None.

**Correctness:**

For someone from a numerical fluid mechanics background, Fig. 7b, 7c, and 7d do not look smooth. As a non-expert in FDTD simulations and nanophotonic structures, I would have appreciated a discussion (Supplementary Material?) on the quality of these results. In other words, are all simulations sufficiently resolved spatially to achieve grid independence? There is no mention of that in the paper. Otherwise, I did not identify issues.

**Documentation:**

I did not identify issues.

**Ethics:**

I did not identify issues.

**Limitations:**

Limitations are discussed in supplementary material.

**Opportunities For Improvement:**

See discussion on Correctness and Relation to Prior Work.

The overall computational cost could have been provided.
Another proofread would be useful to correct remaining typos and grammatical errors.
The text could be more efficient in the first two sections.
The first paragraph of the Intro repeats many sentences from the abstract. I suggest avoiding.


**Relation To Prior Work:**

The relation to previous work on FDTD simulations of similar structures and datasets/benchmarks directly related to nanophotonic structures is unclear. I could only find one reference to FDTD and previous work on benchmarks cited does not include anything directly related to nanophotonic structures, from a quick look at the cited papers. From reading this paper, without specific background in the field of nanophotonic structures, it is therefore unclear how novel is the work presented.

**Summary And Contributions:**

Maxwell's equations are solved through FDTD simulations for a range of nanophotonic structures inspired by real-world applications. A parametric space is covered for each structure, with over 2 million simulations in total. Objective functions for these structures are presented (e.g., maximize absorption in the material) and the performance of several optimization algorithms over the parametric space is evaluated.

---

> ### Author Response · Authors · 2023-08-15
> **Response to Reviewer wCnZ**
>
> We thank the reviewer for your valuable comments.
>
> > The overall computational cost could have been provided.
>
> We provide the overall computational costs here.  To compute wall-clock time for a single simulation, we calculated the average of the elapsed times of 50 randomly selected simulations.  In addition, to compute wall-clock time for all simulations, we multiplied the average of the elapsed times by the number of simulations shown in *Table 2.*
>
> | Structure | Low fidelity, Single (sec.) | Low fidelity, All (hours) | Medium fidelity, Single (sec.) | Medium fidelity, All (hours) | High fidelity, Single (sec.) | High fidelity, All (hours) |
> | --- | --- | --- | --- | --- | --- | --- |
> | Three-layer | 40.1762 | 1663 | 329.4078 | 13639 | 1339.3777 | 55457 |
> | Nanocones | 22.8788 | 254 | 165.5078 | 1839 | 443.2220 | 4925 |
> | Nanospheres | 193.0568 | 3083 | 1414.7214 | 22593 | 2762.6305 | 44118 |
> | Double-sided | 43.8043 | 23374 | 357.7745 | 190912 | 1195.5680 | 637967 |
>
>
> > Another proofread would be useful to correct remaining typos and grammatical errors. The text could be more efficient in the first two sections. The first paragraph of the Intro repeats many sentences from the abstract. I suggest avoiding.
>
> Thank you for pointing this out.  We have carefully revised our submission.  Please see the revised version.
>
>
> > For someone from a numerical fluid mechanics background, Fig. 7b, 7c, and 7d do not look smooth. I would have appreciated a discussion (Supplementary Material?) on the quality of these results.
>
> Similar results were reported in the previous work [R1, R2].  Also, these results are related to simulation fidelity.  We have provided low-, medium-, and high-fidelity evaluations by controlling simulation resolution; please see *Figure 5.*  Higher fidelity yields smoother results.
>
> [R1] B. Wang and P. W. Leu. Tunable and selective resonant absorption in vertical nanowires. Optics Letters, 37(18):3756–3758, 2012.
>
> [R2] B. Wang and P. W. Leu. High index of refraction nanosphere coatings for light trapping in crystalline silicon thin film solar cells. Nano Energy, 13:226–232, 2015.
>
>
> > Are all simulations sufficiently resolved spatially to achieve grid independence?
>
> Yes, all simulations have been sufficiently resolved spatially to achieve grid independence.  We provided three resolutions, low fidelity, medium fidelity, and high fidelity, which are obtained by controlling grid sizes.  The convergence of simulations was confirmed on the high fidelity simulations.
>
>
> > The relation to previous work on FDTD simulations of similar structures and datasets/benchmarks directly related to nanophotonic structures is unclear.
>
> As described in *Section 2.2,* the structures similar to the structures investigated in our project have been studied in recent work.  We implemented such structures in our framework using an abstract class, which is defined in *base_structure.py.*  Our paper offers an easy-to-use framework for machine learning researchers who want to solve real-world problems in parametric structure design.
>
> Moreover, researchers can easily implement new structures with the *BaseStructure* class.  We showcased simple new structures like not close-packed nanospheres and not close-packed nanocones in the updated repository.  These structures are simple extensions of nanocone and nanosphere systems with not close-packed structures.
>
>
> > Without specific background in the field of nanophotonic structures, it is therefore unclear how novel is the work presented.
>
> As described in the **Reviewer MbyG**'s comment *As far as I am aware there is no comparable dataset/benchmark already,* our work is novel.  Our datasets and benchmarks are capable of reducing a gap between the materials science and machine learning fields, by providing easy-to-use simulations and their results.  We have provided attractive nanophotonic structures for diverse real-world tasks and a mold for new structures.  We believe that our study encourages many machine learning researchers to tackle real-world problems in materials science with minimal effort.

---

> > ### Author Response · Authors · 2023-08-23
> >
> > We thank you for your constructive comments.
> >
> > We answered your concerns in the rebuttal and also revised our submission.  We believe that our revision and answers can resolve your concerns.
> >
> > If you have further concerns, please let us know.

---

> > > ### Comment · Reviewer_wCnZ · 2023-08-29
> > >
> > > Thank you for addressing my comments. I will maintain my score at 7, keeping in mind my confidence level of 1.

---

> > > > ### Author Response · Authors · 2023-08-31
> > > >
> > > > Thank you for responding to our response!
> > > >
> > > > We are glad that our response addressed your concerns.

---

### Official Review · Reviewer_eNci · 2023-07-26
**This work proposes benchmarks and systems for examining the optical properties of photonic structures and their parametric designs.**

**Rating:** 4
**Confidence:** 3
**Correctness:** Appear to be appropriate.
**Clarity:** Appropriate

**Strengths:**

The paper focus on simulating the several nanophotonic structures, and developed a simulation pipeline for the structure, and consequently provide the data and optimization of the design parameters. It is a comprehensive framework.

**Additional Feedback:**

It can be an interesting work, but it seems fit better to some computational material journal/conference.

**Documentation:**

Appropriate.

**Limitations:**


There authors have not mentioned much on the limitation.


**Opportunities For Improvement:**

The simulation pipeline may need some validation and calibration. It is not clear how accurate of the established simulation framework. Without proper calibration, the simulation pipeline may not very useful.

It also needs more clarification how the developed simulation compared with other related work such as Meep. What is the advantage of the proposed simulation framework, both in the computation and accuracy, when comparing with other related simulation tools.


**Relation To Prior Work:**

There are some descriptions on the related work.

**Summary And Contributions:**

It is an interesting framework on the electrodynamic simulations. My major concern is that the manuscript may not fit well for this conference. It fits better to conference in computational materials.

---

> ### Author Response · Authors · 2023-08-15
> **Response to Reviewer eNci**
>
> We thank the reviewer for your valuable comments.
>
> > The simulation pipeline may need some validation and calibration. It is not clear how accurate of the established simulation framework. Without proper calibration, the simulation pipeline may not very useful.
>
> Our work does not propose a new simulation tool.  Our work provides a new dataset and benchmark for nanophotonic structures and their parametric design.  Diverse optimization techniques can be employed to find a novel structure defined on a parametric search space.  Since our dataset provides all simulation results, researchers can assess their algorithms without running complex simulations.
>
>
> > It also needs more clarification how the developed simulation compared with other related work such as Meep.
>
> We have used Meep to model and simulate nanophotonic structures defined in our paper.  The datasets and benchmarks are provided using Meep and we furthermore provide code and documentation that enables extensification of the dataset.
>
>
> > It can be an interesting work, but it seems fit better to some computational material journal/conference.
>
> We appreciate the reviewer's acknowledgment that the paper may be of interest.  Regarding the concern about its fit for a computatational material journal/conference, we would like to emphasize that our work is aligned with the themes of the NeurIPS Datasets and Benchmarks Track.  Our research is not solely focused on materials science, but explores the intersection of machine learning, optimziation, and physical systems.
>
> By defining a parametric space of nanophotonic structures, we provide a platform that allows machine learning researchers to access real-world materials science problems with minimal effort.  Our paper also builds upon previous works [R1, R2] in the previous NeurIPS Datasets and Benchmarks Tracks that have successfully bridged physics-informed modeling and artificial intelligence.
>
> We believe in the integrative nature of our work and its potential impact on both the materials science and machine learning communities that make it an appropriate fit for this conference.
>
> [R1] Y. Deng et al. Benchmarking data-driven surrogate simulators for artificial electromagnetic materials. NeurIPS Datasets and Benchmarks Track, 2021.
>
> [R2] A. Thangamuthu et al. Unravelling the performance of physics-informed graph neural networks for dynamical systems. NeurIPS Datasets and Benchmarks Track, 2022.

---

> > ### Author Response · Authors · 2023-08-23
> >
> > We thank you for your constructive comments.
> >
> > We answered your concerns in the rebuttal and also revised our submission.  We believe that our revision and answers can resolve your concerns.
> >
> > If you have further concerns, please let us know.

---

### Official Review · Reviewer_s1HW · 2023-07-27
**Too much focus on electrodynamics/applications background and not enough on the contributed dataset and benchmarks**

**Rating:** 4
**Confidence:** 3

**Strengths:**

The biggest strength of this paper is that a very large number of geometric configurations have been simulated for a number of metal/oxide nanophotonics structures with relevance to important materials and energy applications.  The repository posted by the authors is well-structured and the source code is well documented.

**Additional Feedback:**

n/a

**Clarity:**

The paper is generally well written, however there is an imbalanced focus on material which I believe to be unimportant to the scope of the Datasets and Benchmarking track at NeurIPS.  For example, I don't think there is much value added by including the differential equations solved in Equations (4) and (5) in the main paper.  This has almost no bearing on the quality and significance of the benchmarks and datasets. On the other hand, the authors choose to relegate a clear description of their dataset to sections S.1 and S.3 of the supplementary material.  To me, it should be the other way around.  Additionally, as discussed above, too much time is spent discussing background of electromagnetism and optical simulations.  What is actually contributed by Figure 1 depicting an EM wave?  Much better in my opinion would be to combine Figures 2 and 3 into a single figure. Including a picture of the EM wave but not S.1 or S.3 in the main text seems like an enormous oversight. Finally, I think there is some repetition in the text as well as in the figures, e.g., those depicting the geometry, which could be condensed into a single figure, e.g., three rows and four columns depicting 2D, 3D, and sandwich film structure.

**Correctness:**

I am not able to assess the accuracy of the electrodynamics simulations.  The dataset appears to be constructed in a sound way and a large number of geometrical variations are considered.  I think there would be more value in including a larger variety of materials and geometries and reducing the number of geometrical iterations.  For example, for thin films I am surprised that only TiO2 was considered.  Additionally, in the nanocone and nanosphere systems, it would be interesting to see results for cases where the spheres and cones are not close-packed and the degree of optical coupling could be evaluated.

Finally, I have a major point of contention with the author's statement in the abstract and in the main paper that '... computations can be made ab initio, that is, without assumptions or simplifications'.  All computer simulations and the physical calculations they are based on involve a model which includes at least some assumptions (e.g., those listed in Section 3.3 for your results).  I am not convinced that significant value is being added by this statement, and it is not correct; please do not include it.

**Documentation:**

As discussed, there is not a compelling description of how the dataset would be utilized by other researchers.  The repository linked to the submission is clear and the source code is well-written and well documented.  However, there is no maintenance plan, and there is no discussion as to whether the database will be continuously updated, or whether it will be allowed to be contributed to by other researchers.  As such, the impact is limited.

**Ethics:**

I do not suspect any ethical concerns.

**Limitations:**

There are no significant negative societal impacts from this work

**Opportunities For Improvement:**

Please see summary and contributions section as well as clarity section.

**Relation To Prior Work:**

There is not a clear discussion as to how the authors paper relates to previous contributions in the field of nanophotonics simulation databases and benchmarking.  There was a paper in 2021 in this NeurIPS track on the topic and it is missing from the cited literature (https://openreview.net/pdf?id=-or413Lh_aF). This is just one example, but in general the authors need to explicitly describe previous contributions to dataset + benchmarking for nanophotonics research and how this paper contributes to gaps in the field.  As such, understanding the scope of the impact of the submitted paper is limited.

**Summary And Contributions:**

The aim of the authors paper is to contribute datasets and benchmarks for simulations of nanophotonics structures.  The authors focus on structures which are relevant to solar cells, EM shielding, and anti-reflective coatings. Nanophotonics simulations is an important and difficult area of research, ultimately however I believe this paper as written lies below the acceptance threshold for publication in the NeurIPS Datasets and Benchmarks paper track.

The impact of the dataset is limited and it's unclear to what extent their repository/framework would be useful to other researchers in the field.  A discussion of the relationship to prior work on datasets for nanophotonics simulation is missing.  I am unclear what is specifically contributed in the way of benchmarking and am not confident that I agree with the author's statement that they 'propose benchmarks and systems for assessing the optical properties of photonic structures...'.  The section on benchmarks in the main paper is roughly 1 paragraph and defers the reader to the supplementary material, but there is not enough detail here either to understand what benchmarks are being proposed for evaluating the optical simulations, only the RMSE of their simulations is quoted for various geometric/material configurations. It would be good to see other relevant metrics as well, e.g., core-hours or some measure of computational intensity.

The main results I could identify in the paper involve (1) optical simulation (i.e. transmittance, absorption, and reflectance values, and E/B fields) on the structures presented in Figures 4, 5, and 6 of the paper as a function of various geometric parameters (e.g., thickness) or material choices (e.g., Au or Ag metal); and (2) a comparison of five objective functions for optimization problems on four of the nanostructures. The main strength of the paper is the sheer number of simulations performed.  However, the authors do not provide a compelling account of how the data would be used by other researchers.  Additionally, there is no discussion as to whether the dataset/framework would be open for submission of data from other researchers in the field or whether the repository would be updated later with new information.  There is no maintenance plan.

---

> ### Author Response · Authors · 2023-08-15
> **Response to Reviewer s1HW (1/n)**
>
> We thank the reviewer for your valuable comments.
>
> >  It would be good to see other relevant metrics as well, e.g., core-hours or some measure of computational intensity.
>
> We provide computational intensity here.  To compute wall-clock time for a single simulation, we calculated the average of the elapsed times of 50 randomly selected simulations.  In addition, to compute wall-clock time for all simulations, we multiplied the average of the elapsed times by the number of simulations shown in *Table 2.*
>
> | Structure | Low fidelity, Single (sec.) | Low fidelity, All (hours) | Medium fidelity, Single (sec.) | Medium fidelity, All (hours) | High fidelity, Single (sec.) | High fidelity, All (hours) |
> | --- | --- | --- | --- | --- | --- | --- |
> | Three-layer | 40.1762 | 1663 | 329.4078 | 13639 | 1339.3777 | 55457 |
> | Nanocones | 22.8788 | 254 | 165.5078 | 1839 | 443.2220 | 4925 |
> | Nanospheres | 193.0568 | 3083 | 1414.7214 | 22593 | 2762.6305 | 44118 |
> | Double-sided | 43.8043 | 23374 | 357.7745 | 190912 | 1195.5680 | 637967 |
>
>
> > I think there would be more value in including a larger variety of materials and geometries and reducing the number of geometrical iterations.
>
> We have defined an abstract class in *base_structure.py,* to help define new structures easily.  All the structures implemented in our project inherit from this class.  We think that the use of the abstract class allows us to add new structures and new materials easily.  In the following answers, we provide some new cases.
>
> > For example, for thin films I am surprised that only TiO2 was considered.
>
> Since we conducted all simulations for all combinations of materials, we did not choose some materials like crystalline silicon (cSi), zinc oxide (ZnO), indium tin oxide (ITO), and aluminum-doped zinc oxide (AZO).
>
> Eventually, we included cSi, ZnO, ITO, and AZO along with TiO2 for the three-layer film and double-sided nanocone system, and the simulation results will be able to be shared before publishing our work.  Please see *Section 3.2* and *Table 1* of the revision.
>
>
> > It would be interesting to see results for cases where the spheres and cones are not close-packed and the degree of optical coupling could be evaluated.
>
> By considering your comment, we showed simple new structures like not close-packed nanospheres and not close-packed nanocones.  It is easily implemented with the *BaseStructure* class defined in *base_structure.py.*
>
> Please see the updated repository.
>
>
> > I have a major point of contention with the author's statement in the abstract and in the main paper that '... computations can be made ab initio, that is, without assumptions or simplifications'. ... please do not include it.
>
> Thank you for pointing this out.  We have removed the statement in the revision.  The sentence now states, "Electrodynamic simulations, which are based on Maxwell’s equations, make accurate predictions of the optical and electromagnetic properties of these structures [18]."
>
> [18] D. J. Griffiths. Introduction to electrodynamics. American Association of Physics Teachers, 2005.
>
> > There is an imbalanced focus on material which I believe to be unimportant to the scope of the Datasets and Benchmarking track at NeurIPS. … I don't think there is much value added by including the differential equations solved in Equations (4) and (5) in the main paper.
>
> We have revised our paper thoroughly based on your comment.
>
> > The authors choose to relegate a clear description of their dataset to sections S.1 and S.3 of the supplementary material.
>
> We moved *Sections S.1 and S.3* to the main article.  Please see the revision.
>
> > Additionally, as discussed above, too much time is spent discussing background of electromagnetism and optical simulations. What is actually contributed by Figure 1 depicting an EM wave? Much better in my opinion would be to combine Figures 2 and 3 into a single figure.
>
> We moved *Figure 1* to the supplementary material and combined *Figures 2 and 3.*  In addition, we revised our paper substantially.  Please see the revision; *Figures 2 and 3* are now *Figures 1 and 2.*

---

> ### Author Response · Authors · 2023-08-15
> **Response to Reviewer s1HW (2/n)**
>
> > I think there is some repetition in the text as well as in the figures, e.g., those depicting the geometry, which could be condensed into a single figure, e.g., three rows and four columns depicting 2D, 3D, and sandwich film structure.
>
> Thank you for your suggestion. We have updated Figure 3 based on your suggestions and removed any repitition in the text.
>
>
> > There was a paper in 2021 in this NeurIPS track on the topic and it is missing from the cited literature. ... The authors need to explicitly describe previous contributions to dataset + benchmarking for nanophotonics research and how this paper contributes to gaps in the field.
>
> We have added discussion on several related papers including the paper you mentioned; please see *Section 2.3.*
>
> Our dataset + benchmarking for nanophotonics research is capable of reducing a gap between the materials science and machine learning fields, by providing easy-to-use simulations and their results.  As described above, machine learning researchers readily define new structures with the BaseStructure class.
>
> > However, the authors do not provide a compelling account of how the data would be used by other researchers. ... As discussed, there is not a compelling description of how the dataset would be utilized by other researchers.
>
> Our dataset can be used to find an optimal structure with any optimization techniques on a parametric space of nanophotonic structures; some preliminary results are demonstrated in *Section 5.*  Our work can make many researchers in machine learning and optimization access a real-world materials science problem with minimal effort.
>
> In addition, our datasets serve the outputs of the nanophotonic simulations related to electromagnetic waves.  It can be used to directly predict the behavior of the waves.
>
> > There is no maintenance plan, and there is no discussion as to whether the database will be continuously updated, or whether it will be allowed to be contributed to by other researchers.
>
> To promote the open-source activities of other researchers, we have added instructions to contribute to our project for many researchers. Please see *CODE_OF_CONDUCT.md,* *CONTRIBUTING.md,* and *ISSUE_TEMPLATE* in the repository.

---

> > ### Author Response · Authors · 2023-08-23
> >
> > We thank you for your constructive comments.
> >
> > We carefully considered your comments and revised our submission.  We believe that our revision and answers can resolve your concerns.
> >
> > If you have further concerns, please let us know.

---

### Official Review · Reviewer_MbyG · 2023-07-28
**Good benchmark (with potential to be a great benchmark with some extra effort)**

**Rating:** 7
**Confidence:** 4

**Strengths:**

- Interesting and relevant problem domain: designing nanophotonic structures has real-world relevance and is the subject of a lot of research. Personally I always find it amazing that you can manipulate light in unusual ways with nanostructures to create "impossible" optical conditions.
- Simulations seem well set-up as far as I can tell (using an established open source package is a good start). The code used to produce them is well-documented.
- Provided datasets can be very useful for evaluation and for model-based optimization approaches

**Additional Feedback:**

One question: how do you parameterize the different materials? Is it just via their permittivity and permeability?

In general, I think you have taken a hard high-dimensional optimization problem and converted it to an easier low-dimensional optimization problem. If you undo this and transform this into a high-dimensional open-ended optimization problem using a simulator I would be prepared to considerably raise my score 🙂

**Clarity:**

Yes, it is fairly clear. However, I think the relationship between E, B, T, A, R could be described a bit more clearly for non-experts.

**Correctness:**

Yes, as far as I can tell everything is correct (but I am not an expert in photonics)

**Documentation:**

Yes, the dataset is generally well-documented. I could not run the code because anonymous4openscience only allows files to be downloaded one at a time. I would have preferred if the authors had shared the code as a zip file.

**Ethics:**

no concerns

**Limitations:**

Yes, I think the authors were fairly transparent about the limitations of this work

**Opportunities For Improvement:**

- Design problems are arguably set up in a way that limits the potential of machine learning. By reducing the system to a small number of parameters, the design landscape is small and low-dimensional. This is arguably a regime where classical methods would work quite well. However, the intrinsic problem is high dimensional (designing the placement of material in a 3D space). In my opinion, this is where machine learning would have the greatest potential to help, just as it has helped in other high dimensional problems (e.g. image classification / generation). Perhaps machine learning could come up with unintuitive or complicated designs that human designers would not think of, instead of simple shapes like cones?
- Missed opportunity to create an extensible dataset: in line with above, the use of an open-source simulator with well-defined setup and tunable fidelity suggests that users could potentially run the meep simulator themselves to evaluate the utility of arbitrary structures, in addition to the structures given in the datasets. This could potentially support very interesting open-ended design benchmarks for people without the domain knowledge to correctly set up the FDTD simulations themselves, a bit like the dockstring benchmark for molecule design.
- Benchmarks are fairly limited. Only a few methods were tested, and it doesn't look like the hyperparameters were tuned. I think it is the responsibility of papers in the D&B track which suggest new benchmarks to try to gauge how hard the tasks are. This does not necessarily require testing every method which has ever been proposed but should definitely be more than running a handful of methods with default parameter settings.

**Relation To Prior Work:**

As far as I am aware there is no comparable dataset/benchmark already. However, I think the authors should note:

- This paper in TMLR had a nanophotonic optimization task as a benchmark: https://openreview.net/forum?id=tPMQ6Je2rB
- There have been previous papers attempting the design of nanophotonic structures with machine learning (e.g. https://www.degruyter.com/document/doi/10.1515/nanoph-2019-0474/html)
- In general, it would be nice to have some kind of commentary about how this dataset complements other datasets/benchmarks for inverse design for scientific problems (e.g. molecule design, protein sequence design)

**Summary And Contributions:**

This paper provides some datasets and benchmarks for the design of nanophotonic structures (structures meant to manipulate light with material patterns whose size is similar to the wavelength of light). They define 5 different periodic nanostructures parameterized by a small number of variables (widths, heights, spacing, etc) and define meaningful metrics to optimize for each one. They perform simulations using the Meep package to evaluate the metrics for a dense grid of parameters. Using these datasets the authors provide pre-trained regression models to interpolate between them. They perform some experiments evaluating a range of optimization methods (classical + Bayesian optimization), showing differential evolution and BO to generally be the strongest methods.

**EDIT**: after some discussions with the authors I raised my score to 7 to account for improvements to the paper.

---

> ### Author Response · Authors · 2023-08-15
> **Response to Reviewer MbyG (1/n)**
>
> We thank the reviewer for your valuable comments.
>
> > By reducing the system to a small number of parameters, the design landscape is small and low-dimensional. ... However, the intrinsic problem is high dimensional (designing the placement of material in a 3D space). ... Perhaps machine learning could come up with unintuitive or complicated designs that human designers would not think of, instead of simple shapes like cones? ... In general, I think you have taken a hard high-dimensional optimization problem and converted it to an easier low-dimensional optimization problem. If you undo this and transform this into a high-dimensional open-ended optimization problem using a simulator I would be prepared to considerably raise my score.
>
> We are inspired by your suggestion.
> However, changing the problem to a material placement in 3D space problem has major challenges such as (i) the resulting structure may be impossible to fabricate and (ii) it may be extremely computationally expensive to optimize over such a high-dimensional space (see the table shown below for the total compute used for this paper). This is an interesting idea for future research, but outside the scope of this paper, which seeks to provide datasets and benchmarks.
>
> It should be noted in the Thangamuthu et al. paper [R1] only the dynamics of a 3 link pendulum, 5 link spring, 3D solid cube, and gravitational system were studied.
>
> Nevertheless, our framework encourages many researchers to add new structures, which are defined on higher-dimensional spaces.  Please see the answer of the next concern, which hints at the expansion of our work.
>
> [R1] A. Thangamuthu et al. Unravelling the performance of physics-informed graph neural networks for dynamical systems. NeurIPS Datasets and Benchmarks Track, 2022.
>
> | Structure | Low fidelity, Single (sec.) | Low fidelity, All (hours) | Medium fidelity, Single (sec.) | Medium fidelity, All (hours) | High fidelity, Single (sec.) | High fidelity, All (hours) |
> | --- | --- | --- | --- | --- | --- | --- |
> | Three-layer | 40.1762 | 1663 | 329.4078 | 13639 | 1339.3777 | 55457 |
> | Nanocones | 22.8788 | 254 | 165.5078 | 1839 | 443.2220 | 4925 |
> | Nanospheres | 193.0568 | 3083 | 1414.7214 | 22593 | 2762.6305 | 44118 |
> | Double-sided | 43.8043 | 23374 | 357.7745 | 190912 | 1195.5680 | 637967 |
>
> To compute wall-clock time for a single simulation, we calculated the average of the elapsed times of 50 randomly selected simulations.  In addition, to compute wall-clock time for all simulations, we multiplied the average of the elapsed times by the number of simulations shown in *Table 2.*
>
>
> > Missed opportunity to create an extensible dataset: in line with above, the use of an open-source simulator with well-defined setup and tunable fidelity suggests that users could potentially run the meep simulator themselves to evaluate the utility of arbitrary structures, in addition to the structures given in the datasets.
>
> This is an insightful point and aligns with our goals of maximizing accessibility and usability, but also opening the door for innovation and creativity.  We are motivated to explore ways to structure our dataset and tools to be more extensible and user-friendly.  For example, we have defined an abstract class (defined in *base_structure.py*) to help define new structures easily as shown below.
>
> We have additionally provided guidance and support for users who wish to engage with the simulator and expand on this existing dataset. In particular, we have added *CODE_OF_CONDUCT.md,* *CONTRIBUTING.md,* and *ISSUE_TEMPLATE* in the repository, to promote the open-source activities of other researchers.
>
> Moreover, we have included several higher-dimensional optimization problems based on your suggestions; please see the updated repository.  These structures are simple extensions of nanocone and nanosphere systems with not close-packed structures.

---

> ### Author Response · Authors · 2023-08-15
> **Response to Reviewer MbyG (2/n)**
>
> > Benchmarks are fairly limited. Only a few methods were tested, and it doesn't look like the hyperparameters were tuned. I think it is the responsibility of papers in the D&B track which suggest new benchmarks to try to gauge how hard the tasks are.
>
> We have run L-BFGS-B and a truncated Newton algorithm as well as six algorithms presented in the paper (random search, Powell's method, Py-BOBYQA, DIRECT, differential evolution, and Bayesian optimization).  Unlike the other six algorithms, the L-BFGS-B and truncated Newton algorithm failed to optimize our design problems, which makes us exclude them in the paper.
>
> We faithfully set up the six algorithms.  For example, we used GP regression with the Matern 5/2 kernel as a surrogate function and expected improvement as an acquisition function for the Bayesian optimization algorithm.  Also, all kernel hyperparameters are optimized by marginal likelihood maximization.
>
> We have included the implementation of the algorithms tested in order to help many researchers freely access them.  Many researchers can easily compare them to newly developed algorithms.
>
>
> > As far as I am aware there is no comparable dataset/benchmark already. However, I think the authors should note: ...
>
> Thank you for recommending these interesting papers.  We have added discussion on the papers you recommended. Please see *Section 2.3.*
>
> > One question: how do you parameterize the different materials? Is it just via their permittivity and permeability?
>
> Yes, that is correct, a material is parameterized by its permittivity and permeability as described in *Section 2.1.*  It should be noted that both the permittivity and permeability are complex.  Equivalently, the complex refractive index can be used to parameterize materials.

---

> > ### Comment · Reviewer_MbyG · 2023-08-17
> > **Thanks for responding to my review**
> >
> > Thank you for engaging with my review. I appreciate the changes you made, although it does not really impact my assessment of the paper, which is already leaning positive. I will try to discuss this with the more negative reviewers.
> >
> > My thoughts on what you wrote:
> >
> > ### Missed opportunity to create an extensible dataset
> >
> > I understand the difficulties involved in making the task more open-ended (e.g. optimized structures may not be easy to fabricate, simulation time may be expensive). While I agree that it is extra work and not obligatory (i.e. what you've done is fine), it seems a bit ridiculous to say that this suggestion is "outside the scope of this paper, which seeks to provide datasets and benchmarks." A high-dimensional optimization benchmark with corresponding dataset would fit well with the rest of the paper in my opinion: you could present various datasets and benchmarks with varying degrees of constraints.
> >
> > It appears that one concern of the authors is the computational time required to make a corresponding dataset which covers the space, which would indeed grow as the dimensionality of the problem grows. However, I think the error in this reasoning is thinking that you need to provide a dataset which covers the space as part of the benchmark. I think just providing a constrained simulator is fine. A work you could consider looking at is [dockstring](https://pubs.acs.org/doi/full/10.1021/acs.jcim.1c01334), which proposes a benchmark for molecular design. One of their tasks is to choose the top 5000 molecules out of a list of ~1B molecules. Even though the paper does not actually calculate the scores for all 1B molecules, they provide an open-source simulator which allows the user of the benchmark to evaluate any set of 5000 molecules from the 1B, and define metrics which refer only to the scores of the molecules found, and not how the molecules found relate to the global optimum (which is unknown). When I suggested a benchmark with an extensible dataset, this is what I had in mind.
> >
> > In the rest of your response you more or less stated that researchers can PR additional features into your package, suggesting that this could be added in the future. I don't know if you actually believe this will happen, but I certainly don't: given that it is a lot of work, anybody who puts in the work would probably want to publish it separately and create their own package. There is not really any incentive to merge their work into your package: that's just not how academia works. So to me this is not a very satisfying response.
> >
> > In summary, while I don't think this expansion of the work is necessary for acceptance or is a flaw in any way with what the authors did, I am not very satisfied with the claims that it is "out of scope" or "could be added by the community". If the authors choose to expand the work and add this, I would gladly raise my score to 8-9. Otherwise, if you don't want to do it for whatever reason then just say that you won't do it at this time. I would rather you just say this directly than give excuses for not doing it.
> >
> > ### Benchmarking
> >
> > Thank you for clarifying that you did tune the methods. I think this makes the benchmarks stronger.

---

> > > ### Author Response · Authors · 2023-08-18
> > >
> > > Thank you for your thoughtful response.  We are glad that we can discuss an interesting topic with the reviewer.
> > >
> > > > Missed opportunity to create an extensible dataset
> > >
> > > We generally agree with your points on not satisfying with "out of scope" or "could be added by the community."
> > >
> > > We spent some more time thinking about your suggestion.  We will ignore fabrication issues, and add a higher-dimensional problem as shown in Figure 3f of the revision.  It is an extremely challenging problem that is a generalized form of nanophotonic structures.  This structure is defined on pixel grids or voxel grids.  The goal of this structure design is to find an optimal structure by determining presence or absence of materials for all material blocks; you can imagine Minecraft.  For example if we fill voxels to a shape of vertical wires, it is similar to the vertical nanowire, and if a shape of cones is created by filling material blocks with some material (not air), they become nanocones.  Furthermore, for this structure, we allow that each block can have different material.  For example, adjacent blocks can be made of different materials; different colors represent different materials in Figure 3f.  We would like to emphasize that there exist a vast number of possible combinations so that it can open a challenging problem that can find innovative and creative (hardly investigated) nanophotonic structures.  Please see the corresponding paragraph in Section 3.4 of the revision for the details of this structure.

---

> > > > ### Author Response · Authors · 2023-08-23
> > > >
> > > > We thank you for your constructive comments.
> > > >
> > > > We have updated our submission based on your follow-up comment.
> > > >
> > > > If you have further concerns, please let us know.

---

> > > > ### Comment · Reviewer_MbyG · 2023-08-28
> > > > **Good addition, increasing my score slightly**
> > > >
> > > > I am happy to see this addition, even though it appears you do not yet have any results for it. I will increase my score to 7.
> > > >
> > > > Also, I did initiate a private discussion among the reviewers but none of the other reviewers have engaged with it yet.

---

> > > > > ### Author Response · Authors · 2023-08-29
> > > > >
> > > > > Thank you for considering the revision.
> > > > >
> > > > > We hope that the reviewers will have a constructive discussion on our work.

---

### Author Response · Authors · 2023-08-15
**General Comment to All Reviewers**

We thank the reviewers for their generally favorable feedback and constructive comments that have contributed to enhancing our work.  We are encouraged by the following comments from the reviewers:

**Reviewer MbyG** observed *Interesting and relevant problem domain: designing nanophotonic structures has real-world relevance and is the subject of a lot of research. Personally I always find it amazing that you can manipulate light in unusual ways with nanostructures to create "impossible" optical conditions,* *Simulations seem well set-up as far as I can tell (using an established open source package is a good start). The code used to produce them is well-documented,* and *Provided datasets can be very useful for evaluation and for model-based optimization approaches.*

**Reviewer s1HW** mentioned *The biggest strength of this paper is that a very large number of geometric configurations have been simulated for a number of metal/oxide nanophotonics structures with relevance to important materials and energy applications* and *The repository posted by the authors is well-structured and the source code is well documented.*

**Reviewer eNci** succinctly noted *It is a comprehensive framework.*

**Reviewer wCnZ** remarked *The datasets and benchmarks appear to be useful to develop lower fidelity models targeted for the design of nano materials to form desired nanophotonic structures.*

**Reviewer ZH3g** applauded *The paper is responsive to the conference topic, it is well written, the approach is sound, and overall methodology is appropriate* and *This submission merits acceptance.*

Furthermore, there is consensus among the reviewers about the well-documented nature of our repository.  In light of the reviewers' comments, we have revised our manuscript, supplementary material, and the code repository as detailed below.

We have carefully considered all of the feedback provided, and in response, we have made the following revisions to our manuscript to address the reviewers' concerns and suggestions:

* Revised the abstract
* Removed the expression related to "ab initio"
* Moved the figures on electromagnetic waves and the properties of light to the supplementary material (now, Figures s.1 and s.2)
* Reorganized the Background section (Section 2)
* Reorganized and enhanced the Related Work section (Section 2.3)
* Reorganized the Datasets and Benchmarks section (Section 3)
* Updated the figures on nanophotonic structures (now, Figure 2)
* Simplified the Structures of Interest section (Section 3.3)
* Added a new structure, Combinatorial System with Material Blocks, and its description (Figure 2f and Section 3.3)
* Added four materials (cSi, ZnO, ITO, and AZO) for the three-layer films and double-sided nanocone systems
* Improved the Datasets section (Section 3.4)
* Moved the tables on datasets (now, Tables 1 and 2) from the supplementary material
* Updated the figures on dataset visualization (now, Figures 3 and 4)
* Fixed minor issues on grammar and typos.

---

> ### Author Response · Authors · 2023-08-28
>
> Dear reviewers and area chairs,
>
> We sincerely appreciate your effort and time dedicated to our submission.
>
> Since the reviewer-author discussion period is over soon, we would like to ask you if your concerns are fully resolved.
>
> Please let us know if you have further questions.
>
> We hope to communicate with you.  Thank you again!
>
> Best regards,
>
> Authors.

---

### Decision · Program_Chairs · 2023-09-22

**Decision:**

Accept (Poster)

**Comment:**

The paper provides a dataset and benchmark for designing nanophotonic structures via optimization. The reviewers and I are in agreement that these design problems are an impactful use-case for ML in science and provides a reasonable benchmark and baseline results for these problems. These settings have the potential to be further improved by new machine learning methods and will serve as a great reference point for future papers in the area. There were a few concerns raised throughout the reviewing process (e.g., on validation, calibration, baselines, and the positioning of the paper), we recommend the authors to incorporate these into the final version of the paper.